# White Gaussian Noise Constraints for Reward-Guided Generation

## Abstract

We propose a constrained optimization framework that preserves white Gaussian noise characteristics during latent optimization for reward-guided generation. At its core is a novel constraint formulation that allows efficient projection while tightly characterizing white Gaussian noise. In deep generative models, supplying white Gaussian noise as input is essential for stable and realistic generation, but preserving its characteristics during optimization remains challenging. This challenge is amplified in reward-guided generation, where gradient-based updates can exploit the reward and produce unrealistic or low-quality outputs. Prior methods address this by introducing regularization terms that encourage certain white Gaussian noise properties, particularly in the spectral domain. However, regularization offers only soft penalties and cannot guarantee that the latent vector retains the white Gaussian noise characteristics throughout optimization. To overcome this, we propose a constrained optimization approach that directly projects the latent vector onto a feasible set. Leveraging a bijective mapping to a compact spectral domain, we define constraints that tightly characterize white Gaussian noise and induce a feasible set with a closed-form projection, enabling efficient updates through projected gradient ascent. In experiments on reward-guided text-to-image generation, our approach outperforms regularization-based baselines across four reward functions in terms of reward, sample quality, and maximization speed.

## 1 Introduction

At the core of deep generative modeling for high-dimensional continuous data lies the use of the *standard Gaussian distribution* as a canonical latent prior, valued for its simplicity and ease of sampling. Specifically, generative models take a latent vector whose elements are independently drawn from the standard Gaussian distribution—i.e., *white Gaussian noise*—and map it to the data space. Supplying such white Gaussian noise is essential for stable and realistic generation. While sampling the noise is straightforward, *preserving white Gaussian noise characteristics during optimization*, particularly when the latent is updated via gradient-based methods, remains challenging.

The challenge becomes particularly critical in *reward-guided generation*, where the latent vector is optimized to maximize a task-specific reward. While this enables controllable generation, it also introduces failure cases: without explicit enforcement of white Gaussian noise characteristics, the optimization may exploit the reward and produce unrealistic or low-quality samples. To prevent this, it is crucial to ensure that the latent vector preserves the characteristics of white Gaussian noise throughout optimization.

Prior approaches (Eyring et al., 2024; Tang et al., 2024; Hwang et al., 2025) mitigate this issue by incorporating regularization terms designed to preserve certain Gaussian properties. From the perspective of *white Gaussian noise*, characterized by both Gaussian statistics and low spatial correlation, PRNO (Tang et al., 2024) emphasized the importance of reducing spatial correlation, and MPGR (Hwang et al., 2025) showed that spectral domain regularization offers an effective and efficient way to reduce spatial correlation. Motivated by this, we adopt the spectral domain to characterize white Gaussian noise. However, regularization inherently imposes only soft penalties, and thus offers no guarantee that the latent vector will retain the characteristics throughout optimization.

To overcome these limitations, we propose a different approach: a *constrained optimization* framework that enforces white Gaussian noise characteristics by explicitly restricting the latent vector to a feasible set, rather than penalizing deviations from it. This guarantees that the latent vector satisfies the

desired properties of white Gaussian noise. Crucially, this is enabled by a closed-form projection onto the feasible set, which allows efficient updates via **projected gradient ascent**.

To realize this idea, we first construct a compact spectral domain that removes redundancy from the Fourier coefficients of real-valued vectors. A sample from the standard Gaussian distribution in the spatial domain is mapped to a sample from the standard complex Gaussian distribution in this domain. Hence, in this domain, we define **white Gaussian noise constraints** based on blockwise $\ell_1$ and $\ell_2$ norms with known Gaussian expectations. The spectral formulation provides interpretability through its connection to white noise characteristics and spatial decorrelation. The constraints not only **tightly characterize white Gaussian noise**, but also **allow an efficient closed-form projection** onto the feasible set.

In the experiments, we validate our method on test-time latent optimization for one-step text-to-image generative models, such as FLUX-schnell (Labs, 2024) and SDXL-Turbo (Sauer et al., 2024). At each iteration, we replace regularization with a closed-form projection that preserves the white Gaussian structure of the latent and its gradient. We compare against representative regularizers and report both the given reward and held-out human-preference metrics (Aesthetic Score, PickScore, HPSv2, ImageReward) to assess the trade-off. Across all rewards, our approach attains stronger target scores without degrading held-out metrics, yields more stable samples, and converges quickly with negligible overhead and no extra hyperparameters to tune.

## 2 RELATED WORK

**Reward-Guided Generation via Latent Optimization.** Latent optimization, also referred to as *noise optimization*, is a test-time procedure that updates the latent input to a generative model to maximize a task-specific reward. Unlike fine-tuning (Black et al., 2023; Clark et al., 2024; Wallace et al., 2024), this approach avoids model retraining and is therefore computationally efficient. It has been applied across diverse modalities, including images (Eyring et al., 2024; Tang et al., 2024), motion (Karunratanakul et al., 2024; Zhao et al., 2025), and music (Novack et al., 2024). Since continuous generative models take white Gaussian noise as input, it is crucial to regularize the latent vector during optimization so that it retains the properties of white Gaussian noise: (i) Gaussian marginal distribution and (ii) low spatial correlation. Below, we review regularization methods that focus on (i) marginal distribution and (ii) spatial structure together.

**Regularizing Marginal Distribution.** Regularization strategies in machine learning are often designed to enforce statistical properties on model weights during optimization, such as through KL divergence (Kingma & Welling, 2014) or kurtosis (Shkolnik et al., 2020), but these regularization terms can also be incorporated into test-time latent optimization. For reward-guided generation, prior works (Samuel et al., 2023; Ben-Hamu et al., 2024; Eyring et al., 2024) regularize the $\ell_2$ norm of the latent vector to encourage alignment with the statistics of the Gaussian distribution. However, these marginal-based approaches treat the latent as an unordered set and therefore fail to regularize *spatial correlation*, which is essential for preserving the property of white Gaussian noise.

**Regularizing Both Marginal and Spatial Structure.** Structure-aware methods regularize both the marginal distribution and spatial correlation by treating the latent vector as an ordered sequence. For example, PRNO (Tang et al., 2024) penalizes the mean and covariance of latent subvectors, promoting spatial decorrelation and Gaussian statistics. Building on the spectral characterization of white noise via flat power spectra (Khintchine, 1934; Oppenheim et al., 1999; Stoica & Moses, 2005), Hwang et al. (2025) propose a spectral regularizer that penalizes deviations in blockwise $\ell_1$ norms of Fourier coefficients from Gaussian statistics. This spectral domain regularization more effectively reduces spatial correlations and preserves white Gaussian noise characteristics. We therefore characterize white Gaussian noise in the spectral domain. In contrast to prior regularization-based approaches, we impose constraints that capture the properties of white Gaussian noise and enable efficient projection onto the feasible set.

## 3 LATENT OPTIMIZATION: REGULARIZATION VS. CONSTRAINTS

Given a generative model $\mathcal{M} : \mathbb{R}^N \to \mathbb{F}$ that maps a latent vector to the data space $\mathbb{F}$, and a reward function $r : \mathbb{F} \to \mathbb{R}$ that scores generated outputs, the goal is to find $x \in \mathbb{R}^N$ that maximizes the reward. Optimizing the reward alone often causes the latent vector to deviate from white Gaussian

noise characteristics, resulting in unrealistic outputs. To prevent this, a common approach is to add a regularization term $\mathcal{L}_{\text{reg}}$ that encourages $\boldsymbol{x}$ to retain a certain property of white Gaussian noise:

$$\max_{\boldsymbol{x} \in \mathbb{R}^N} r(\mathcal{M}(\boldsymbol{x})) - \lambda \mathcal{L}_{\text{reg}}(\boldsymbol{x}), \tag{1}$$

where $\lambda$ is a hyperparameter. For example, $\mathcal{L}_{\text{reg}}$ may penalize $\ell_2$ norm deviations (Samuel et al., 2023), or blockwise $\ell_1$ norm deviations in the spectral domain (Hwang et al., 2025), relative to Gaussian statistics.

In practice, the optimization is performed via gradient ascent due to the complexity of the generative models. However, it is unclear how effectively $\mathcal{L}_{\text{reg}}$ is minimized during optimization. As a result, regularization may fail to enforce the characteristics of white Gaussian noise, or conversely, may hinder reward maximization.

To address this limitation, we propose replacing regularization with explicit constraints. We directly constrain $\boldsymbol{x}$ to lie in a feasible set $\mathcal{G} \subset \mathbb{R}^N$ that enforces certain white Gaussian noise characteristics:

$$\max_{\boldsymbol{x} \in \mathbb{R}^N} r(\mathcal{M}(\boldsymbol{x})) \quad \text{subject to} \quad \boldsymbol{x} \in \mathcal{G}. \tag{2}$$

This formulation enables the use of *projected gradient ascent*, where each iteration consists of a gradient update followed by projection onto the feasible set $\mathcal{G}$. This ensures that the latent vector maintains the desired properties of white Gaussian noise throughout the optimization.

---
**Algorithm 1** Projected Gradient Ascent

**Iterate:**
    $J \leftarrow r(\mathcal{M}(\boldsymbol{x}))$
    $\boldsymbol{x} \leftarrow \text{Proj}_{\mathcal{G}}\left(\boldsymbol{x} + \eta \, \nabla_{\boldsymbol{x}} J\right)$

---

A key requirement of this approach is the effectiveness of the projection, which must be applied at every iteration. Simply defining $\mathcal{G}$ as the set of minimizers of an existing regularization term $\mathcal{L}_{\text{reg}}$ is often insufficient because such a set may not allow a closed-form or computationally efficient projection, making each optimization step costly or intractable.

Indeed, even if the projection is well defined, it may fail to preserve white Gaussian noise characteristics if $\mathcal{G}$ does not tightly capture its properties. Therefore, the constraints should be designed with two key objectives: **(i) efficient closed-form projection** and **(ii) tight characterization of white Gaussian noise**. These are essential for effectively applying projected gradient ascent to the latent optimization problem. In the following section, we define constraints that satisfy both criteria.

## 4    White Gaussian Noise Constraints

In this section, we define and analyze constraints that tightly characterize white Gaussian noise while still admitting an efficient closed-form projection. We build on a prior work that regularizes in the spectral domain and extend it into a constrained formulation with an explicit projection operator.

**Notation.** Let $N = 2PB$ be the dimension of a latent vector $\boldsymbol{x} \in \mathbb{R}^N$, where $B$ denotes the block size. The $p$-th block subvector is defined as $\boldsymbol{x}^{(p)} = [x_{pB}, \ldots, x_{pB+B-1}]^\top$. Let $\boldsymbol{F} \in \mathbb{C}^{N \times N}$ be the unitary Discrete Fourier Transform (DFT) matrix, and define the DFT coefficients by $\hat{\boldsymbol{x}} = \boldsymbol{F}\boldsymbol{x}$, which can be efficiently computed using the Fast Fourier Transform (FFT) in $\mathcal{O}(N \log N)$ time.

Our starting point is the regularization term proposed by Hwang et al. (2025), which operates in the spectral domain and encourages each block of DFT coefficients to match the statistics of a complex Gaussian distribution. Concretely, they introduce a blockwise $\ell_1$ penalty

$$\mathcal{L}_{\text{power}}(\boldsymbol{x}) = \frac{1}{N} \sum_{p=0}^{2P-1} \left| \left\| \hat{\boldsymbol{x}}^{(p)} \right\|_1 - \mu B \right|, \tag{3}$$

where $\mu = 0.875$ (close to $\frac{\sqrt{\pi}}{2} \approx 0.886$). This design is based on two observations: (i) for Gaussian latent vectors, not only the marginal distribution but also the spatial correlation is important, and (ii) these correlations are handled in the spectral domain, where they appear in the DFT magnitudes. By regularizing blockwise statistics of $\hat{\boldsymbol{x}}$, $\mathcal{L}_{\text{power}}$ captures part of this structure.

However, directly turning $\mathcal{L}_{\text{power}}$ into a hard constraint, i.e., enforcing $\mathcal{L}_{\text{power}}(\boldsymbol{x}) = 0$, leads to a feasible set for which no efficient closed-form projection is known. The main difficulty stems from

the Hermitian symmetry of the DFT coefficients: different blocks of $\hat{\boldsymbol{x}}$ are correlated, and some coefficients are real-valued while others are complex-valued. This symmetry makes it challenging to derive an efficient closed-form projection algorithm.

To overcome this obstacle, we first define a *compact spectral domain* that removes the Hermitian redundancy while preserving all independent degrees of freedom (Section 4.1). This is achieved via a bijective mapping that aggregates the independent DFT coefficients into a complex-valued vector.

On this compact spectral domain, we then define blockwise constraints that follow the spirit of $\mathcal{L}_{\text{power}}$ but are formulated directly in terms of Gaussian statistics (Section 4.2). Using the relation between the spatial and compact spectral domains, we derive an efficient projection algorithm onto the feasible set (Section 4.3). We then relate our constraints to existing regularization-based approaches and show that they provide a tighter characterization of white Gaussian noise (Section 4.4). Finally, we offer complementary perspectives that connect our constraints to the classical notion of white noise and to the reduction of autocorrelation (Section 4.5).

We refer to our constraints as the **white Gaussian noise constraints**, as they enforce the statistical properties of the Gaussian distribution while ensuring that no frequency component dominates, consistently with the notion of white noise.

## 4.1 Bijective Mapping to Compact Spectral Domain

Our first step is to remove the Hermitian redundancy that arises in the DFT of real-valued latent vectors. This redundancy couples different frequency components and complicates the derivation of a closed-form projection. To address this, we construct a compact spectral domain that contains only the independent DFT coefficients while preserving all information in the spatial domain.

Recall that $\hat{\boldsymbol{x}} = \boldsymbol{F}\boldsymbol{x}$ denotes the DFT of $\boldsymbol{x} \in \mathbb{R}^N$ with even $N$. Then, the DFT coefficients $\hat{\boldsymbol{x}}$ satisfy Hermitian symmetry (Lemma 1):

$$\hat{x}_0, \ \hat{x}_{N/2} \in \mathbb{R} \quad \text{and} \quad \hat{x}_k = \overline{\hat{x}_{N-k}} \quad \text{for } k = 1, \dots, \tfrac{N}{2} - 1. \tag{4}$$

Thus, only $N/2$ complex degrees of freedom are independent: $\hat{x}_0$ and $\hat{x}_{N/2}$, and the first half of the nontrivial frequency coefficients $\hat{x}_1, \dots, \hat{x}_{N/2-1}$. The remaining entries are fully determined by conjugate symmetry. This redundancy complicates the direct formulation of constraints on $\hat{\boldsymbol{x}}$, particularly when deriving closed-form projections, since one must explicitly handle these dependencies.

To resolve this, we define the mapping $\mathcal{F} : \mathbb{R}^N \to \mathbb{C}^{N/2}$ as:

$$\boxed{\boldsymbol{y} = \mathcal{F}(\boldsymbol{x}) \quad \Longleftrightarrow \quad y_0 = \tfrac{\hat{x}_0}{\sqrt{2}} + \tfrac{\hat{x}_{N/2}}{\sqrt{2}} i, \quad y_k = \hat{x}_k \ \text{ for } k = 1, \dots, \tfrac{N}{2} - 1.} \tag{5}$$

This mapping eliminates the redundancy in $\hat{\boldsymbol{x}}$ while preserving all independent components. Specifically, it selects the first half of the complex-valued DFT coefficients and combines the two real-valued terms, $\hat{x}_0$ and $\hat{x}_{N/2}$, into a single complex number. Intuitively, $\mathcal{F}$ yields a compact spectral domain representation with no internal redundancy.

Importantly, this mapping satisfies the following desirable properties:

**Theorem 1.** *The mapping $\mathcal{F}$ is a bijection from $\mathbb{R}^N$ to $\mathbb{C}^{N/2}$. Moreover, if $\boldsymbol{z} \sim \mathcal{CN}(\boldsymbol{0}, \boldsymbol{I}_{N/2})$, then $\mathcal{F}^{-1}(\boldsymbol{z}) \sim \mathcal{N}(\boldsymbol{0}, \boldsymbol{I}_N)$.*

*Proof.* See Appendix B. □

By Theorem 1, enforcing that $\boldsymbol{y} = \mathcal{F}(\boldsymbol{x})$ follows the distribution $\mathcal{CN}(\boldsymbol{0}, \boldsymbol{I}_{N/2})$ is equivalent to enforcing that $\boldsymbol{x}$ follows $\mathcal{N}(\boldsymbol{0}, \boldsymbol{I}_N)$. Leveraging this property, we define the spectral domain constraints in the next subsection.

## 4.2 Modeling White Gaussian Noise Constraints

On the compact spectral domain defined by $\mathcal{F}$, we now specify the constraints that define our feasible set. The design follows the *blockwise* spectral regularization $\mathcal{L}_{\text{power}}$ (Equation 3): instead of acting on the entire spectrum at once, we partition the DFT coefficients into local blocks of size $B$ and constrain their aggregate statistics.

Compared to imposing only *global* constraints (e.g., matching the total $\ell_1$ and $\ell_2$ norms of the entire vector), the blockwise constraints define a strictly smaller feasible set. Global constraints fix only two scalar quantities for all coefficients combined, so many different configurations remain admissible. In contrast, our formulation fixes the $\ell_1$ and $\ell_2$ norms *in every block*, introducing $2P$ equality constraints instead of just two. As a result, the feasible set under blockwise constraints is a proper subset of the set defined by the corresponding global constraints, and therefore provides a tighter characterization.

Assume $\boldsymbol{y} \sim \mathcal{CN}(\boldsymbol{0}, \boldsymbol{I}_{N/2})$, which is equivalent to each element $y_j$ being an i.i.d. sample from $\mathcal{CN}(0,1)$. Then any block $\boldsymbol{y}^{(p)}$ of size $B$ has the expected blockwise norms

$$\mathbb{E}\left[\|\boldsymbol{y}^{(p)}\|_1\right] = \tfrac{\sqrt{\pi}}{2}B, \qquad \mathbb{E}\left[\|\boldsymbol{y}^{(p)}\|_2^2\right] = B. \tag{6}$$

Using these theoretical statistics of $\mathcal{CN}(0,1)$, we impose **blockwise $\ell_1$ and $\ell_2$ norm constraints in the compact spectral domain**, requiring that within every block the norms exactly match their theoretical expectations under $\mathcal{CN}(0,1)$. We use the same block size $B = 16$ as Hwang et al. (2025). These constraints define the compact spectral domain feasible set

$$\mathcal{G}_{\mathbb{C}} = \left\{ \boldsymbol{y} \in \mathbb{C}^{N/2} \; : \; \|\boldsymbol{y}^{(p)}\|_1 = \tfrac{\sqrt{\pi}}{2}B, \quad \|\boldsymbol{y}^{(p)}\|_2 = \sqrt{B}, \quad p = 0, \ldots, P-1 \right\}. \tag{7}$$

Finally, we lift the definition back to the spatial domain.

$$\boxed{\mathcal{G}_{\mathbb{R}} = \left\{ \boldsymbol{x} \in \mathbb{R}^N : \mathcal{F}(\boldsymbol{x}) \in \mathcal{G}_{\mathbb{C}} \right\}} \tag{8}$$

We refer to the blockwise $\ell_1$ and $\ell_2$ norm constraints in the compact spectral domain as the **white Gaussian noise constraints**. They not only follow the Gaussian statistics but also prevent any single frequency component from dominating in magnitude, thereby aligning with the notion of white noise. Correspondingly, we call the spatial feasible set $\mathcal{G}_{\mathbb{R}}$ the **white Gaussian noise feasible set**. A more detailed interpretation from the perspective of white noise is provided in Section 4.5.

### 4.3 PROJECTION ONTO WHITE GAUSSIAN NOISE FEASIBLE SET

In this section, we introduce the closed-form projection onto the feasible set $\mathcal{G}_{\mathbb{R}}$ defined in equation 8. Specifically, given an input $\boldsymbol{x} \in \mathbb{R}^N$, we aim to find its projection $\dot{\boldsymbol{x}} \in \mathcal{G}_{\mathbb{R}}$ such that:

$$\dot{\boldsymbol{x}} = \operatorname*{argmin}_{\tilde{\boldsymbol{x}} \in \mathbb{R}^N} \|\boldsymbol{x} - \tilde{\boldsymbol{x}}\|_2^2 \quad \text{subject to} \quad \tilde{\boldsymbol{x}} \in \mathcal{G}_{\mathbb{R}}. \tag{9}$$

Since $\mathcal{G}_{\mathbb{R}}$ is defined via spectral constraints, the projection is naturally formulated in the compact spectral domain. Letting $\boldsymbol{y} = \mathcal{F}(\boldsymbol{x})$ and $\dot{\boldsymbol{y}} = \mathcal{F}(\dot{\boldsymbol{x}})$, we obtain:

$$\dot{\boldsymbol{y}} = \operatorname*{argmin}_{\tilde{\boldsymbol{y}} \in \mathbb{C}^{N/2}} \|\mathcal{F}^{-1}(\boldsymbol{y}) - \mathcal{F}^{-1}(\tilde{\boldsymbol{y}})\|_2^2 \quad \text{subject to} \quad \tilde{\boldsymbol{y}} \in \mathcal{G}_{\mathbb{C}}. \tag{10}$$

To simplify this problem, we can utilize the following property of the mapping $\mathcal{F}^{-1}$:

**Theorem 2.** *The mapping $\mathcal{F}^{-1}$ is $\mathbb{R}$-linear, and for any $\boldsymbol{z} \in \mathbb{C}^{N/2}$, $\|\mathcal{F}^{-1}(\boldsymbol{z})\|_2^2 = 2\|\boldsymbol{z}\|_2^2$.*

*Proof.* See Appendix C. $\qquad\square$

By Theorem 2, we obtain $\|\mathcal{F}^{-1}(\boldsymbol{y}) - \mathcal{F}^{-1}(\tilde{\boldsymbol{y}})\|_2^2 = \|\mathcal{F}^{-1}(\boldsymbol{y} - \tilde{\boldsymbol{y}})\|_2^2 = 2\|\boldsymbol{y} - \tilde{\boldsymbol{y}}\|_2^2$.

Thus, the original projection problem reduces to:

$$\dot{\boldsymbol{y}} = \operatorname*{argmin}_{\tilde{\boldsymbol{y}} \in \mathbb{C}^{N/2}} \|\boldsymbol{y} - \tilde{\boldsymbol{y}}\|_2^2 \quad \text{subject to} \quad \tilde{\boldsymbol{y}} \in \mathcal{G}_{\mathbb{C}}. \tag{11}$$

This corresponds to projecting onto the set $\mathcal{G}_{\mathbb{C}}$, and the projection can be performed independently on each block. In other words, the projection onto $\mathcal{G}_{\mathbb{R}}$ in the spatial domain reduces to a projection onto $\mathcal{G}_{\mathbb{C}}$ in the compact spectral domain, which further decomposes into blockwise projections onto the intersection of $\ell_1$ and $\ell_2$ spheres. The closed-form projection onto this intersection are known to exist (Liu et al., 2019). We rigorously derive the projection algorithm for $\mathcal{G}_{\mathbb{C}}$ in Appendix D. Here, we summarize the resulting algorithm.

For the $p$-th block, $\boldsymbol{y}^{(p)}$, consisting of the values $y_{pB}, \ldots, y_{pB+B-1}$, we define $\boldsymbol{w}$ as the descending-sorted array:

$$\boldsymbol{w} = \texttt{sort\_descending}\left(\{|y_{pB}|, \ldots, |y_{pB+B-1}|\}\right). \tag{12}$$

We then define the cumulative sums $S_{1,k} = \sum_{l=0}^{k} w_l$ and $S_{2,k} = \sum_{l=0}^{k} w_l^2$.

Then, for $k+1 \geq \frac{\pi}{4}B$, there exists a unique $k^*$ satisfying the following condition (define $w_B = -\infty$):

$$w_{k^*+1} \leq \lambda^{(k^*)} < w_{k^*}, \quad \text{where } \lambda^{(k^*)} = \frac{S_{1,k^*}}{k^*+1} - \frac{\sqrt{\pi}}{2} \frac{\sqrt{B}}{k^*+1} \sqrt{\frac{(k^*+1)S_{2,k^*} - S_{1,k^*}^2}{k^*+1-\frac{\pi}{4}B}}. \tag{13}$$

The final solution is then given by:

$$\dot{y}_j = \frac{\sqrt{\pi}}{2} B \frac{\text{ReLU}\left(|y_j| - \lambda^{(k^*)}\right)}{S_{1,k^*} - (k^*+1)\lambda^{(k^*)}} \frac{y_j}{|y_j|} \qquad \text{for } j = pB, \ldots, pB + B - 1. \tag{14}$$

The projection onto $\mathcal{G}_{\mathbb{C}}$ takes $\mathcal{O}(N \log B)$ time. See Appendix D.1 for a detailed explanation.

To summarize, the projection onto $\mathcal{G}_{\mathbb{R}}$ involves computing $\boldsymbol{y} = \mathcal{F}(\boldsymbol{x})$, projecting onto $\mathcal{G}_{\mathbb{C}}$ to obtain $\dot{\boldsymbol{y}}$, and recovering $\dot{\boldsymbol{x}} = \mathcal{F}^{-1}(\dot{\boldsymbol{y}})$. Since $\mathcal{F}$ and its inverse are implemented via the Fast Fourier Transform (FFT), the total procedure takes $\mathcal{O}(N \log N)$ time, which accounts for only 0.04% of the runtime in our experiments with FLUX model.

Using this projection, we generated 1M samples of $\boldsymbol{x} \sim \mathcal{N}(\boldsymbol{0}, \boldsymbol{I}_N)$ with $N = 65536$, and computed the cosine similarity between $\boldsymbol{x}$ and its projection onto $\mathcal{G}_{\mathbb{R}}$. The results showed a minimum cosine similarity of 0.988, indicating that white Gaussian noise is very close to the feasible set $\mathcal{G}_{\mathbb{R}}$.

## 4.4 CONNECTIONS TO PRIOR REGULARIZATION METHODS

We explain the connection between our constraints and prior regularization terms—$\ell_2$-norm regularization and spectral domain regularization—and show that our blockwise $\ell_2$ and $\ell_1$ norm constraints are analogous to the minimization of each, respectively. This reveals that our constraints provide a tighter characterization of white Gaussian noise while also enabling efficient closed-form projection.

$\ell_2$**-Norm Regularization (Samuel et al., 2023).** $\ell_2$-norm regularization aims to maximize the likelihood of the Euclidean norm $\|\boldsymbol{x}\|_2$ under the assumption that $\boldsymbol{x} \sim \mathcal{N}(\boldsymbol{0}, \boldsymbol{I}_N)$, where $\|\boldsymbol{x}\|_2 \sim \chi_N$. The corresponding loss is

$$\mathcal{L}_{\text{norm}}(\boldsymbol{x}) = -\log p_{\chi_N}(\|\boldsymbol{x}\|_2), \quad p_{\chi_N}(r) = \frac{1}{2^{N/2-1}\Gamma(N/2)} r^{N-1} e^{-r^2/2}, \quad r \geq 0. \tag{15}$$

This loss is minimized when $\|\boldsymbol{x}\|_2^2 = N-1$, which corresponds to vectors lying on a hypersphere. For high-dimensional generative models such as FLUX ($N = 65536$) and SDXL-Turbo ($N = 16384$), this is nearly equal to the expectation $\mathbb{E}[\|\boldsymbol{x}\|_2^2] = N$.

Our $\ell_2$-norm constraints, $\|\boldsymbol{y}^{(p)}\|_2 = \sqrt{B}$, closely approximate the minimum of $\mathcal{L}_{\text{norm}}$, given that

$$\|\boldsymbol{x}\|_2^2 = \|\mathcal{F}^{-1}(\boldsymbol{y})\|_2^2 = 2\|\boldsymbol{y}\|_2^2 = 2 \sum_{p=0}^{P-1} \|\boldsymbol{y}^{(p)}\|_2^2 \tag{16}$$

by Theorem 2. Thus, $\boldsymbol{x} \in \mathcal{G}_{\mathbb{R}}$ ensures $\|\boldsymbol{x}\|_2^2 = 2PB = N$ which differs only slightly from the optimal value $N-1$. Although our constraints are imposed in the spectral domain, they inherently influence the spatial domain as well.

The key distinction is that $\ell_2$-norm regularization, while easily minimized via scaling, does not constrain spatial correlations (Figure 1). In contrast, our formulation enforces local spectral structure, resulting in a tighter characterization of white Gaussian noise.

**Spectral Domain Regularization (Hwang et al., 2025).** The spectral domain regularization term, $\mathcal{L}_{\text{power}}$ (Equation 3), penalizes deviations in the blockwise $\ell_1$ norms of the DFT coefficients.

This regularization exploits the statistical properties of the complex Gaussian distribution $\mathcal{CN}(\boldsymbol{0}, \boldsymbol{I}_B)$ and is similar to our approach in its use of blockwise $\ell_1$ norms. However, our formulation further incorporates blockwise $\ell_2$ constraints, providing a tighter characterization of white Gaussian noise.

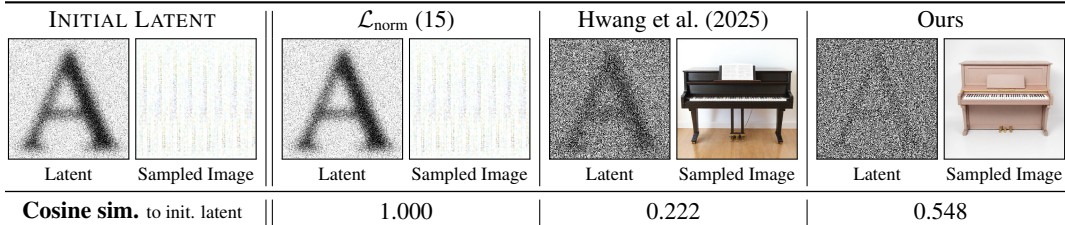

| INITIAL LATENT | | $\mathcal{L}_{\text{norm}}$ (15) | | Hwang et al. (2025) | | Ours | |
|---|---|---|---|---|---|---|---|
| Latent | Sampled Image | Latent | Sampled Image | Latent | Sampled Image | Latent | Sampled Image |
| **Cosine sim.** to init. latent || | 1.000 || | 0.222 || | 0.548 || |

Figure 1: **Effectiveness of Projection onto** $\mathcal{G}_{\mathbb{R}}$. Starting from an initial latent encoding the letter 'A', we compare two regularization methods and our projection. Our method preserves high cosine similarity to the initial latent while reducing the spatial correlations. Unlike Hwang et al. (2025), which requires slow gradient-based iterative projection, our method guarantees optimality with a single operation. The images are sampled from FLUX with the prompt "Piano".

The key distinction between Hwang et al. (2025) and our approach lies in the spectral representation. While $\mathcal{L}_{\text{power}}$ directly regularizes the DFT coefficients $\hat{x} = Fx$, we instead use the compact representation $y = \mathcal{F}(x)$. This choice allows us to directly define a tractable feasible set $\mathcal{G}_{\mathbb{C}}$ and derive an efficient projection even with tighter characterization of white Gaussian noise. As a result, projection onto the minimizer set of $\mathcal{L}_{\text{power}}$ requires gradient descent and is significantly slower and less accurate. In contrast, our formulation achieves projection with a single operation, avoiding heavy and imprecise additional computations, as shown in Figure 1.

### 4.5 INTERPRETATION OF THE CONSTRAINTS IN DIFFERENT ASPECTS

Although we have so far explained our constraints using properties of the Gaussian distribution, they can also be interpreted from several other perspectives. We summarize the interpretations below.

**White Noise Perspective.** The constraints prevent any single DFT coefficient from becoming dominant at a specific frequency by limiting the total budget within each block. For example, when $B = 16$, the theoretical maximum of $|y_j|^2$ is approximately 7.18 (see Appendix E), whereas the total budget is $N/2$ for typical $N \gg 10^4$. This ensures that no individual frequency can disproportionately dominate the spectrum. As a result, the constraints prevent the signal from exhibiting strong and localized patterns—closely mimicking the characteristics of white noise, in which all samples are independently drawn and no strong structured signal is present.

**Autocorrelation Reduction Perspective.** Our constraints assign equal magnitude budgets across blocks that are ordered by frequency. Although $\mathcal{F}$ only retains the first half of the DFT coefficients, the Hermitian symmetry (Lemma 1) ensures that the remaining half is constrained as well. This leads to a spread of DFT magnitudes across frequencies, while still permitting some variance.

From the perspective of circular autocorrelation, defined as:

$$r_{x}[\ell] \;=\; \frac{1}{N} \sum_{n=0}^{N-1} x_n \, x_{n-\ell \;(\text{mod } N)}, \qquad \ell = 0, \ldots, N-1, \tag{17}$$

the squared magnitudes of the DFT coefficients form a DFT pair with the autocorrelation:

$$r_{x}[\ell] = \frac{1}{N} \sum_{k=0}^{N-1} |\hat{x}_k|^2 \, e^{2\pi i k \ell / N}, \quad \ell = 0, \ldots, N-1. \tag{18}$$

This implies that spreading the DFT magnitudes reduces spatial autocorrelation. A more detailed discussion is provided in Appendix F.

## 5 LATENT OPTIMIZATION WITH TEXT-TO-IMAGE GENERATIVE MODELS

Building on prior work such as ReNO (Eyring et al., 2024) and MPGR (Hwang et al., 2025), we present experimental results on latent optimization for one-step text-to-image generation.

**Reward Models.** We evaluate using four human-preference–based reward models: *Aesthetic Score* (Schuhmann et al., 2022), a predictor of visual appeal trained on human–rated images;

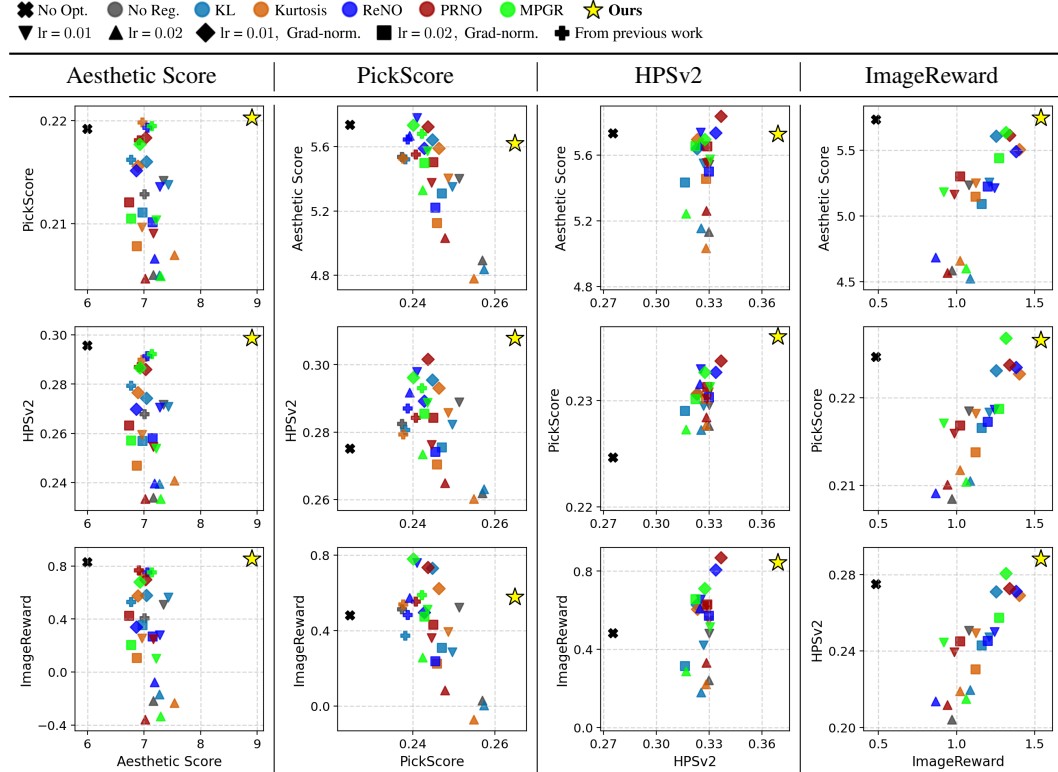

Figure 2: **Quantitative results with FLUX-schnell.** Each column corresponds to the same given reward (x-axis), and different held-out rewards (y-axis). Each point denotes the score after 200 iterations, with higher positions and more rightward placement indicating better trade-offs. For baselines, multiple points are plotted across learning rates and regularization schemes. Our method consistently achieves the best trade-off across all reward–held-out reward pairs.

*PickScore* (Kirstain et al., 2023), trained using paired human judgments of text–image outputs; *HPSv2* (Wu et al., 2023), a human preference predictor correlating strongly with subjective evaluations; and *ImageReward* (Xu et al., 2023), trained on expert annotations for text-to-image evaluation. When one of these models is used as the given reward, the others serve as held-out rewards to assess generalization and determine whether the effects transfer across different human-aligned criteria.

**Baselines.** We use prompts from the animal dataset (Black et al., 2023) for Aesthetic Score and 60 prompts randomly sampled from T2I-Compbench++ (Huang et al., 2025) for the other reward models. We report results using FLUX-schnell (Labs, 2024) as the generative model, with additional experiments on SDXL-Turbo (Sauer et al., 2024) provided in Appendix J. We compare our constrained optimization method against several regularization-based approaches: KL (Kingma & Welling, 2014), Kurtosis (Shkolnik et al., 2020), ReNO (Eyring et al., 2024), PRNO (Tang et al., 2024), and MPGR (Hwang et al., 2025). Two reference baselines are also included: one without optimization (No Opt.) and one without regularization (No Reg.). In addition, we evaluate two weighting schemes: (i) fixed-weight regularization, where the regularization gradient is scaled only by the coefficient $\lambda$, and (ii) gradient-normalized regularization, where the regularization gradient is rescaled to match the magnitude of the reward gradient to ensure balanced contributions. For Aesthetic Score and PickScore, we also report the results from Hwang et al. (2025).

**Implementation Details.** We initialize each run from white Gaussian noise and optimize for 200 iterations with FLUX and 50 iterations with SDXL-Turbo. The optimization is performed using Adam (Kingma & Ba, 2014) optimizer, with gradient clipping at 0.03 and learning rates of 0.02 (FLUX) and 0.1 (SDXL-Turbo). For all regularization-based methods, we also report results with learning rates 0.01 and 0.05, respectively. We set the regularization coefficient to 2.0 for the baselines. Our method projects both the latent vector and its gradient onto the white Gaussian noise feasible set at every iteration. All experiments were conducted on a single NVIDIA A6000 GPU, taking approximately 1 minute (FLUX) and 20 seconds (SDXL-Turbo) per 50 iterations, respectively.

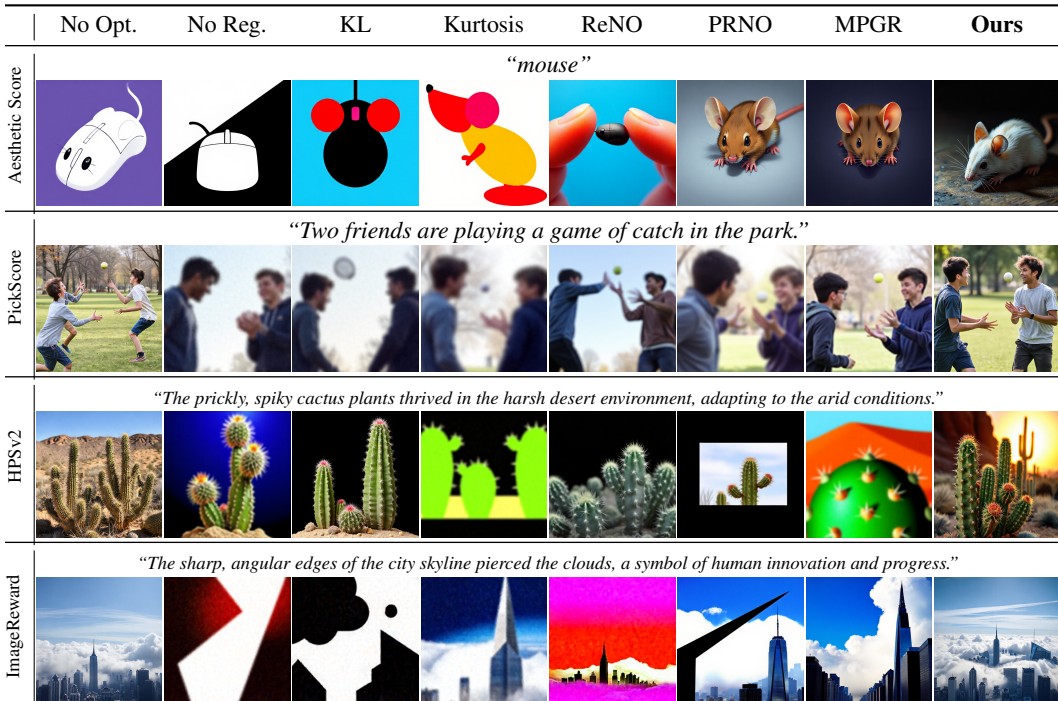

Figure 3: **Qualitative results with FLUX-schnell.** Columns denote optimization method; rows correspond to the given reward, with the prompt shown above each row. Our constrained optimization preserves realism and prompt fidelity while attaining higher target scores and strong held-out quality.

**Results.** Quantitative and qualitative results are shown in Figures 2 and 3. According to Figure 2, on *Aesthetic Score*, baselines raise the given reward from 6 to at most 7.5, whereas ours reaches around 9; *PickScore*, *HPSv2*, and *ImageReward* show similar behavior, with our method forming the rightmost frontier. Across all settings, we increase the *given* reward substantially *without* significantly losing *held-out* rewards compared to the unoptimized (No Opt.) reference, indicating improved alignment without degrading realism or image quality. In terms of reward maximization speed, our method reaches a comparable Aesthetic Score using only 30% of the iterations (see Appendix H).

Looking at the qualitative results in Figure 3, regularization-based methods such as PRNO and MPGR, which penalize spatial correlations, often yield realistic outputs. However, this effect is inconsistent, as seen in the second and third examples in Figure 3. These observations highlight the lack of principled design in soft regularization and reinforce the advantage of our constrained optimization. Our method reliably produces realistic images across all reward models and, as illustrated in the fourth row, aligns more faithfully with the text prompt. For example, it explicitly renders the "skyline pierced the clouds", which is not achieved by the unoptimized reference.

## 6 CONCLUSION

We addressed the latent optimization problem, where preserving the properties of white Gaussian noise is essential. Our solution is a constrained optimization framework that enforces white Gaussian noise characteristics via lightweight projection onto a feasible set at each iteration. We designed constraints that tightly characterize white Gaussian noise and admit a closed-form projection, formulated in the spectral domain for both efficiency and interpretability in terms of white noise and spatial decorrelation. In the experiments, our approach achieves reliable reward maximization without sacrificing sample quality.

ETHICS STATEMENT

We affirm compliance with the ICLR Code of Ethics. The work is conducted solely with publicly available models and datasets, without human subjects, user data, or personally identifiable information. We note the potential for misuse of generative AI and recommend responsible deployment.

REPRODUCIBILITY STATEMENT

We plan to release the code upon publication. Proofs of theorems and detailed derivations are provided in the appendix.

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

## A   Hermitian Symmetry

**Lemma 1.** *Let $\boldsymbol{x} \in \mathbb{R}^N$, and let $\hat{\boldsymbol{x}} = \boldsymbol{F}\boldsymbol{x}$ denote its DFT. Then, $\hat{\boldsymbol{x}}$ satisfies the Hermitian symmetry (Proakis, 2007):*

$$\hat{x}_k = \overline{\hat{x}_{N-k}} \quad for \ k = 0, \dots, N-1, \tag{19}$$

*and the coefficients $\hat{x}_0$ and $\hat{x}_{N/2}$ are real-valued.*

*Proof.*

$$\hat{x}_k = \frac{1}{\sqrt{N}} \sum_{j=0}^{N-1} x_j e^{-2\pi \frac{jk}{N} i} = \overline{\frac{1}{\sqrt{N}} \sum_{j=0}^{N-1} x_j e^{2\pi \frac{jk}{N} i}} = \overline{\frac{1}{\sqrt{N}} \sum_{j=0}^{N-1} x_j e^{-2\pi \frac{-jk}{N} i}} = \overline{\hat{x}_{-k \bmod N}} \tag{20}$$

In particular, $\hat{x}_0 = \overline{\hat{x}_0}$ and $\hat{x}_{N/2} = \overline{\hat{x}_{N/2}}$ are real valued. $\qquad \square$

## B   Proof of Theorem 1

**Theorem 1.** *The mapping $\mathcal{F}$ is a bijection from $\mathbb{R}^N$ to $\mathbb{C}^{N/2}$. Moreover, if $\boldsymbol{z} \sim \mathcal{CN}(\boldsymbol{0}, \boldsymbol{I}_{N/2})$, then $\mathcal{F}^{-1}(\boldsymbol{z}) \sim \mathcal{N}(\boldsymbol{0}, \boldsymbol{I}_N)$.*

*Proof.* Let $N$ be even and let $\boldsymbol{F} \in \mathbb{C}^{N \times N}$ be the unitary DFT. For $\boldsymbol{x} \in \mathbb{R}^N$ write $\hat{\boldsymbol{x}} = \boldsymbol{F}\boldsymbol{x}$ and define $\mathcal{F} : \mathbb{R}^N \to \mathbb{C}^{N/2}$ by

$$y_0 = \frac{\hat{x}_0}{\sqrt{2}} + \frac{\hat{x}_{N/2}}{\sqrt{2}} i, \qquad y_k = \hat{x}_k, \quad k = 1, \dots, \tfrac{N}{2} - 1. \tag{21}$$

**Bijectivity.**  Given $\boldsymbol{y} \in \mathbb{C}^{N/2}$, define $\hat{\boldsymbol{x}} \in \mathbb{C}^N$ by

$$\hat{x}_0 = \sqrt{2}\,\Re(y_0), \quad \hat{x}_{N/2} = \sqrt{2}\,\Im(y_0), \quad \hat{x}_k = y_k, \quad \hat{x}_{N-k} = \overline{y_k} \quad (k = 1, \dots, \tfrac{N}{2} - 1), \tag{22}$$

and put

$$\boldsymbol{x} = \boldsymbol{F}^\dagger \hat{\boldsymbol{x}} \in \mathbb{R}^N, \tag{23}$$

where $\boldsymbol{F}^\dagger$ denotes the conjugate transpose of $\boldsymbol{F}$. Since $\boldsymbol{F}$ is unitary, we have $\boldsymbol{F}^\dagger = \boldsymbol{F}^{-1}$.

Equation 22 enforces Hermitian symmetry of $\hat{\boldsymbol{x}}$, whence $\boldsymbol{x}$ in equation 23 is real. It is immediate from equation 21–equation 22 that this construction inverts equation 21; hence $\mathcal{F}^{-1}$ exists and is given by equation 22–equation 23. Therefore $\mathcal{F}$ is a bijection.

**Gaussian Preservation.** Let $\boldsymbol{z} \sim \mathcal{CN}(\boldsymbol{0}, \boldsymbol{I}_{N/2})$ with independent elements, construct $\hat{\boldsymbol{x}} \in \mathbb{C}^N$ by

$$\hat{x}_0 = \sqrt{2}\,\Re(z_0), \quad \hat{x}_{N/2} = \sqrt{2}\,\Im(z_0), \quad \hat{x}_k = z_k, \quad \hat{x}_{N-k} = \overline{z_k} \quad (k = 1, \dots, \tfrac{N}{2} - 1), \tag{24}$$

and set $\boldsymbol{x} = \boldsymbol{F}^\dagger \hat{\boldsymbol{x}} \in \mathbb{R}^N$. Since $\Re(z_0), \Im(z_0) \overset{\text{i.i.d.}}{\sim} \mathcal{N}(0, \tfrac{1}{2})$, $E[z_k] = 0$, $E[z_k \overline{z_\ell}] = \delta_{k\ell}$, and $E[z_k^2] = 0$, $\hat{\boldsymbol{x}}$ is jointly Gaussian and Hermitian-symmetric. We verify the spectral covariance element-wise: variances

$$\mathbb{E}[\hat{x}_0 \overline{\hat{x}_0}] = 1, \qquad \mathbb{E}[\hat{x}_{N/2} \overline{\hat{x}_{N/2}}] = 1, \qquad \mathbb{E}[\hat{x}_k \overline{\hat{x}_k}] = \mathbb{E}[\hat{x}_{N-k} \overline{\hat{x}_{N-k}}] = 1, \tag{25}$$

off-diagonals across different indices vanish by independence, within each conjugate pair $\{k, N-k\}$

$$\mathbb{E}[\hat{x}_k \, \overline{\hat{x}_{N-k}}] = \mathbb{E}[z_k^2] = 0, \tag{26}$$

and within the real coefficients is zero by independence,

$$\mathbb{E}[\hat{x}_0 \, \overline{\hat{x}_{N/2}}] = \mathbb{E}[\hat{x}_{N/2} \, \overline{\hat{x}_0}] = 0, \tag{27}$$

as are all real-complex cross terms with $k \in \{1, \dots, \tfrac{N}{2} - 1\}$. Hence

$$\mathbb{E}[\hat{\boldsymbol{x}}\hat{\boldsymbol{x}}^\dagger] = \boldsymbol{I}_N. \tag{28}$$

Unitarity of $\boldsymbol{F}$ gives

$$\mathrm{Cov}(\boldsymbol{x}) = \mathbb{E}[\boldsymbol{x}\boldsymbol{x}^\top] = \Re(\mathbb{E}[\boldsymbol{x}\boldsymbol{x}^\dagger]) = \Re(\boldsymbol{F}^\dagger \mathbb{E}[\hat{\boldsymbol{x}}\hat{\boldsymbol{x}}^\dagger] \boldsymbol{F}) = \Re(\boldsymbol{F}^\dagger \boldsymbol{I}_N \boldsymbol{F}) = \boldsymbol{I}_N. \tag{29}$$

Therefore, $\boldsymbol{x}$ is a real jointly Gaussian vector with diagonal covariance $\boldsymbol{I}_N$. For *real* jointly Gaussian vectors, a diagonal covariance implies independence. Hence, the elements of $\boldsymbol{x}$ are i.i.d. $\mathcal{N}(0,1)$. Equivalently, $\mathcal{F}^{-1}(\boldsymbol{z}) \sim \mathcal{N}(\boldsymbol{0}, \boldsymbol{I}_N)$ with independent elements.

$\qquad \square$

## C Proof of Theorem 2

**Theorem 2.** *The mapping $\mathcal{F}^{-1}$ is $\mathbb{R}$-linear, and for any $\boldsymbol{z} \in \mathbb{C}^{N/2}$, $\|\mathcal{F}^{-1}(\boldsymbol{z})\|_2^2 = 2\|\boldsymbol{z}\|_2^2$.*

*Proof.* To show the $\mathbb{R}$-linearity of $\mathcal{F}^{-1}$, we first show the $\mathbb{R}$-linearity of $\mathcal{F}$.

**$\mathbb{R}$-linearity of $\mathcal{F}$.** Let $\boldsymbol{x}, \boldsymbol{y} \in \mathbb{R}^N$ and $a, b \in \mathbb{R}$. Write $\hat{\boldsymbol{x}} = \boldsymbol{F}\boldsymbol{x}$ and $\hat{\boldsymbol{y}} = \boldsymbol{F}\boldsymbol{y}$. By linearity of the DFT,

$$\boldsymbol{F}(a\boldsymbol{x} + b\boldsymbol{y}) = a\,\boldsymbol{F}\boldsymbol{x} + b\,\boldsymbol{F}\boldsymbol{y}. \tag{30}$$

Recall the definition of $\mathcal{F}$ (equation 5): for $k = 1, \ldots, \frac{N}{2} - 1$, $(\mathcal{F}(\boldsymbol{x}))_k = \hat{x}_k$, $(\mathcal{F}(\boldsymbol{x}))_0 = \frac{1}{\sqrt{2}}\hat{x}_0 + \frac{1}{\sqrt{2}}\hat{x}_{N/2}\,i$, and analogously for $\boldsymbol{y}$. Each component of $\mathcal{F}$ is a $\mathbb{R}$-linear combination of elements of $\hat{\boldsymbol{x}}$ (or $\hat{\boldsymbol{y}}$). Hence

$$\mathcal{F}(a\boldsymbol{x} + b\boldsymbol{y}) = a\,\mathcal{F}(\boldsymbol{x}) + b\,\mathcal{F}(\boldsymbol{y}). \tag{31}$$

Therefore $\mathcal{F}$ is $\mathbb{R}$-linear.

**$\mathbb{R}$-linearity of $\mathcal{F}^{-1}$.** Since $\mathcal{F}$ is bijective (Theorem 1), for arbitrary $\boldsymbol{u}, \boldsymbol{v} \in \mathbb{C}^{N/2}$ there exist unique $\boldsymbol{x}, \boldsymbol{y} \in \mathbb{R}^N$ with $\boldsymbol{u} = \mathcal{F}(\boldsymbol{x})$ and $\boldsymbol{v} = \mathcal{F}(\boldsymbol{y})$. Using the $\mathbb{R}$-linearity of $\mathcal{F}$,

$$\boldsymbol{u} + \boldsymbol{v} = \mathcal{F}(\boldsymbol{x}) + \mathcal{F}(\boldsymbol{y}) = \mathcal{F}(\boldsymbol{x} + \boldsymbol{y}), \qquad a\boldsymbol{u} + b\boldsymbol{v} = \mathcal{F}(a\boldsymbol{x} + b\boldsymbol{y}), \tag{32}$$

for all $a, b \in \mathbb{R}$. Apply $\mathcal{F}^{-1}$ to both equalities to obtain

$$\mathcal{F}^{-1}(a\boldsymbol{u} + b\boldsymbol{v}) = \mathcal{F}^{-1}\big(\mathcal{F}(a\boldsymbol{x} + b\boldsymbol{y})\big) = a\,\boldsymbol{x} + b\,\boldsymbol{y} = a\,\mathcal{F}^{-1}(\boldsymbol{u}) + b\,\mathcal{F}^{-1}(\boldsymbol{v}). \tag{33}$$

Thus $\mathcal{F}^{-1}$ is $\mathbb{R}$-linear.

**Norm Relation.** We first introduce Parseval's identity:

$$\|\boldsymbol{x}\|_2^2 = \|\hat{\boldsymbol{x}}\|_2^2. \tag{34}$$

*Proof of equation 34.* Let $\boldsymbol{F} \in \mathbb{C}^{N \times N}$ be the unitary DFT, so $\boldsymbol{F}^\dagger \boldsymbol{F} = \boldsymbol{I}_N$. Then

$$\|\hat{\boldsymbol{x}}\|_2^2 = \hat{\boldsymbol{x}}^\dagger \hat{\boldsymbol{x}} = (\boldsymbol{F}\boldsymbol{x})^\dagger (\boldsymbol{F}\boldsymbol{x}) = \boldsymbol{x}^\dagger \boldsymbol{F}^\dagger \boldsymbol{F}\boldsymbol{x} = \boldsymbol{x}^\dagger \boldsymbol{x} = \|\boldsymbol{x}\|_2^2. \tag{35}$$

This proves equation 34.

By the definition of $\mathcal{F}$ (cf. equation 5), write $\boldsymbol{y} = \mathcal{F}(\boldsymbol{x})$ from $\hat{\boldsymbol{x}}$ as

$$y_0 = \frac{\hat{x}_0}{\sqrt{2}} + \frac{\hat{x}_{N/2}}{\sqrt{2}}\,i, \qquad y_k = \hat{x}_k \quad (k = 1, \ldots, \tfrac{N}{2} - 1). \tag{36}$$

We now connect $\|\boldsymbol{y}\|_2^2$ and $\|\boldsymbol{x}\|_2^2$ step by step. First, expand $\|\hat{\boldsymbol{x}}\|_2^2$ by pairing the conjugate-symmetric bins:

$$\|\hat{\boldsymbol{x}}\|_2^2 = |\hat{x}_0|^2 + |\hat{x}_{N/2}|^2 + \sum_{k=1}^{N/2-1} \left( |\hat{x}_k|^2 + |\hat{x}_{N-k}|^2 \right). \tag{37}$$

Using equation 36,

$$|y_0|^2 = \left| \tfrac{1}{\sqrt{2}}\hat{x}_0 + \tfrac{1}{\sqrt{2}}\hat{x}_{N/2}\,i \right|^2 = \tfrac{1}{2}|\hat{x}_0|^2 + \tfrac{1}{2}|\hat{x}_{N/2}|^2 \implies |\hat{x}_0|^2 + |\hat{x}_{N/2}|^2 = 2\,|y_0|^2. \tag{38}$$

For $k = 1, \ldots, \frac{N}{2} - 1$, we have $y_k = \hat{x}_k$, and by Hermitian symmetry $|\hat{x}_{N-k}| = |\hat{x}_k|$, hence

$$|\hat{x}_k|^2 + |\hat{x}_{N-k}|^2 = 2\,|y_k|^2 \qquad (k = 1, \ldots, \tfrac{N}{2} - 1). \tag{39}$$

Substituting equation 38 and equation 39 into equation 37 yields

$$\|\hat{\boldsymbol{x}}\|_2^2 = 2\,|y_0|^2 + \sum_{k=1}^{N/2-1} 2\,|y_k|^2 = 2\,\|\boldsymbol{y}\|_2^2. \tag{40}$$

Combining equation 34 and equation 40 gives

$$\|\boldsymbol{x}\|_2^2 = \|\hat{\boldsymbol{x}}\|_2^2 = 2\,\|\boldsymbol{y}\|_2^2 \iff \|\boldsymbol{y}\|_2^2 = \tfrac{1}{2}\,\|\boldsymbol{x}\|_2^2. \tag{41}$$

**Corollary (Inverse map).** Let $\boldsymbol{z} \in \mathbb{C}^{N/2}$ and set $\boldsymbol{x} = \mathcal{F}^{-1}(\boldsymbol{z})$. Then $\boldsymbol{z} = \mathcal{F}(\boldsymbol{x})$ and equation 41 with $\boldsymbol{y} = \boldsymbol{z}$ gives

$$\|\boldsymbol{x}\|_2^2 = 2\,\|\boldsymbol{z}\|_2^2 \quad \text{i.e.,} \quad \|\mathcal{F}^{-1}(\boldsymbol{z})\|_2^2 = 2\,\|\boldsymbol{z}\|_2^2. \tag{42}$$

$\square$

## D  PROJECTION ONTO THE WHITE GAUSSIAN NOISE FEASIBLE SET IN THE SPECTRAL DOMAIN

We propose an $\mathcal{O}(N \log B)$ algorithm ($N = 2PB$) that computes the closest vector $\dot{\boldsymbol{y}} \in \mathcal{G}_{\mathbb{C}}$ (equation 7) to a given input $\boldsymbol{y} \in \mathbb{C}^{N/2}$ in Euclidean distance.

As edge cases, situations may arise where the projection solution is not uniquely defined. The first occurs when more than $78.5\%$ of the elements in $\boldsymbol{y}^{(p)}$ (i.e., $\frac{\pi}{4}B$ entries) have *exactly equal* magnitudes, and that shared magnitude is the largest within the block. The second occurs when an element has *exactly zero* magnitude. Although both scenarios are exceedingly rare under 64-bit floating-point arithmetic—unless artificially constructed—we address them by perturbing each such $y_j$ with a small complex Gaussian noise $\delta\epsilon$, where $\delta = 10^{-6}$ and $\epsilon \sim \mathcal{CN}(0,1)$, thereby ensuring that the projection remains well-defined and robust.

Let us now derive the closest vector $\dot{\boldsymbol{y}} \in \mathcal{G}_{\mathbb{C}}$ mathematically.

Given an input $\boldsymbol{y} \in \mathbb{C}^{N/2}$, we aim to solve the following optimization problem:

$$\dot{\boldsymbol{y}} = \underset{\tilde{\boldsymbol{y}} \in \mathbb{C}^{N/2}}{\operatorname{argmin}} \|\tilde{\boldsymbol{y}} - \boldsymbol{y}\|_2^2$$

$$\text{subject to} \quad \|\tilde{\boldsymbol{y}}^{(p)}\|_1 = \tfrac{\sqrt{\pi}}{2}B, \quad \|\tilde{\boldsymbol{y}}^{(p)}\|_2^2 = B, \quad \text{for all } p = 0, \ldots, P-1,$$

where $\dot{\boldsymbol{y}}^{(p)} \in \mathbb{C}^B$ denotes the $p$-th block sub-vector of $\dot{\boldsymbol{y}}$.

To solve this problem, we introduce the following Lagrangian function associated with the constraints:

$$\mathcal{L}(\dot{\boldsymbol{y}}, \lambda_1, \lambda_2) = \frac{1}{2} \|\dot{\boldsymbol{y}} - \boldsymbol{y}\|_2^2 + \sum_{p=0}^{P-1} \lambda_{1,p} \left( \left\|\dot{\boldsymbol{y}}^{(p)}\right\|_1 - \tfrac{\sqrt{\pi}}{2}B \right) + \sum_{p=0}^{P-1} \lambda_{2,p} \left( \left\|\dot{\boldsymbol{y}}^{(p)}\right\|_2^2 - B \right), \quad (43)$$

To compute the optimality condition, we differentiate the Lagrangian with respect to $\dot{\boldsymbol{y}}$. Let $\boldsymbol{I}_p \in \mathbb{C}^{\frac{N}{2} \times \frac{N}{2}}$ denote a diagonal *block indicator matrix* that selects the $p$-th block of $\dot{\boldsymbol{y}}$, i.e., $[\boldsymbol{I}_p]_{jj} = 1$ if $j \in \{pB, \ldots, pB + B - 1\}$ and 0 otherwise. Then, the gradient of the Lagrangian becomes:

$$\nabla_{\dot{\boldsymbol{y}}} \mathcal{L}(\dot{\boldsymbol{y}}, \lambda_1, \lambda_2) = \dot{\boldsymbol{y}} - \boldsymbol{y} + \sum_{p=0}^{P-1} \lambda_{1,p} \boldsymbol{I}_p \nabla_{\dot{\boldsymbol{y}}} \|\dot{\boldsymbol{y}}\|_1 + \sum_{p=0}^{P-1} \lambda_{2,p} \boldsymbol{I}_p \dot{\boldsymbol{y}}. \quad (44)$$

To derive the optimality condition, we set the gradient of the Lagrangian to zero, i.e., $\nabla_{\dot{\boldsymbol{y}}} \mathcal{L}(\dot{\boldsymbol{y}}, \lambda_1, \lambda_2) = 0$. Let us now focus on the $j$-th coordinate of the $p$-th block, i.e., for indices $j \in \{pB, \ldots, pB + B - 1\}$. The first-order condition for each such $j$ becomes:

$$(1 + \lambda_{2,p}) \dot{y}_j + \lambda_{1,p} \nabla_{\dot{y}_j} |\dot{y}_j| = y_j. \quad (45)$$

If $\dot{y}_j \neq 0$, the subgradient of the complex $\ell_1$-norm simplifies to:

$$\nabla_{\dot{y}_j} |\dot{y}_j| = \frac{\dot{y}_j}{|\dot{y}_j|}, \quad (46)$$

so the optimality condition becomes:

$$(1 + \lambda_{2,p}) \dot{y}_j + \lambda_{1,p} \frac{\dot{y}_j}{|\dot{y}_j|} = y_j. \quad (47)$$

Since the Lagrange multipliers $\lambda_{1,p}$ and $\lambda_{2,p}$ are real-valued, each optimal solution $\dot{y}_j$ can be expressed in polar form as $\dot{y}_j = s_j e^{i\theta_j}$, where $s_j \geq 0$ denotes the magnitude and $\theta_j$ the phase. If $y_j \neq 0$, we set $\theta_j = \arg(y_j)$, as the optimality condition equation 47 implies that $\dot{y}_j$ and $y_j$ must share the same phase. If $y_j = 0$, then $\theta_j$ can be chosen arbitrarily, since the direction is unconstrained in this case.

To justify that $s_j \geq 0$ holds at the optimum, suppose by contradiction that the optimal solution is $\dot{y}_j = -s_j e^{i\theta_j}$ for some $s_j > 0$. Then, flipping the sign yields:

$$\left| y_j - (-s_j e^{i\theta_j}) \right|^2 = \left| y_j + s_j e^{i\theta_j} \right|^2 < \left| y_j - s_j e^{i\theta_j} \right|^2, \quad (48)$$

which contradicts the assumption that $\dot{y}_j = -s_j e^{i\theta_j}$ is optimal, since both solutions have the same magnitude and thus satisfy the constraints equally well. Therefore, the optimal $s_j$ must be non-negative.

Substituting $\dot{y}_j = s_j e^{i\theta_j}$ into the equation 47 yields:

$$\left(1 + \lambda_{2,p}\right) s_j + \lambda_{1,p} = |y_j|. \tag{49}$$

Otherwise, $\dot{y}_j = 0$, we can still express it in polar form as $\dot{y}_j = s_j e^{i\theta_j}$ with $s_j = 0$ and arbitrary $\theta_j$.

Thus, regardless of whether $\dot{y}_j$ is zero or nonzero, we may uniformly express the solution in polar form as $\dot{y}_j = s_j e^{i\theta_j}$ with $s_j \geq 0$. This reformulation allows us to convert the original complex-valued projection problem into a real-valued optimization over the magnitudes $s_j$. For each block indexed by $p$, we solve:

$$\min_{\{s_j \geq 0\}} \sum_{j=pB}^{pB+B-1} \left(s_j - |y_j|\right)^2 \quad \text{subject to} \quad \sum_{j=pB}^{pB+B-1} s_j = \frac{\sqrt{\pi}}{2}B, \quad \sum_{j=pB}^{pB+B-1} s_j^2 = B. \tag{50}$$

To solve this constrained optimization problem, we introduce the Karush–Kuhn–Tucker (KKT) conditions. The corresponding Lagrangian is defined as:

$$\mathcal{L}'(s, \lambda_1, \lambda_2, \boldsymbol{\tau}) = \sum_{j=pB}^{pB+B-1} \tfrac{1}{2}\left(s_j - |y_j|\right)^2$$
$$+ \lambda_1 \left(\sum_{j=pB}^{pB+B-1} s_j - \frac{\sqrt{\pi}}{2}B\right) + \lambda_2 \left(\sum_{j=pB}^{pB+B-1} s_j^2 - B\right) + \sum_{j=pB}^{pB+B-1} \tau_j s_j, \tag{51}$$

where $\tau_j \leq 0$ are the KKT multipliers associated with the non-negativity constraints $s_j \geq 0$.

Differentiating with respect to $s_j$ for $j = pB, \ldots, pB + B - 1$, and applying the complementary slackness condition yields:

$$s_j - |y_j| + \lambda_1 + 2\lambda_2 s_j + \tau_j = 0, \quad \tau_j \leq 0, \quad \tau_j s_j = 0. \tag{52}$$

First, consider the case $\tau_j < 0$. Then, by complementary slackness, it must be that $s_j = 0$. Substituting into the stationarity condition gives:

$$\tau_j = |y_j| - \lambda_1 < 0, \tag{53}$$

which implies $|y_j| < \lambda_1$.

On the other hand, if $\tau_j = 0$, then the stationarity condition simplifies to:

$$(1 + 2\lambda_2)s_j = |y_j| - \lambda_1, \quad s_j \geq 0. \tag{54}$$

Next, we analyze the structure of the optimal solution. Since the constraints enforce both $\sum_j s_j = \frac{\sqrt{\pi}}{2}B$ and $\sum_j s_j^2 = B$, at least two distinct $s_j$'s must be strictly positive.

Also, suppose that there exist two indices $j_1, j_2 \in \{pB, \ldots, pB + B - 1\}$ such that $s_{j_1} > s_{j_2}$ but $|y_{j_1}| < |y_{j_2}|$. Consider swapping their values while preserving all constraints. The change in the objective value would be:

$$(s_{j_1} - |y_{j_1}|)^2 + (s_{j_2} - |y_{j_2}|)^2 > (s_{j_2} - |y_{j_1}|)^2 + (s_{j_1} - |y_{j_2}|)^2,$$

since the squared error decreases when the larger $s_j$ is matched with the larger $|y_j|$. This contradicts optimality. Hence, the optimal values $s_j$ must preserve the same ordering as the magnitudes $|y_j|$ within each block.

Furthermore, the norm constraints imply that at least two of the $s_j$ values must differ. Let $s_{j_1} > s_{j_2} > 0$, then from the optimality condition:

$$(1 + \lambda_2)(s_{j_1} - s_{j_2}) = (|y_{j_1}| - \lambda_1) - (|y_{j_2}| - \lambda_1) = |y_{j_1}| - |y_{j_2}| \geq 0.$$

This implies $1 + \lambda_2 \geq 0$, and thus

$$(1 + 2\lambda_2)s_j = \max\{|y_j| - \lambda_1,\, 0\} = \mathrm{ReLU}\left(|y_j| - \lambda_1\right), \tag{55}$$

where we used the fact that $s_j = 0$ precisely when $|y_j| < \lambda_1$. This compact expression captures both KKT branches in a unified form.

If $1 + 2\lambda_2 = 0$, then the stationarity condition simplifies to:

$$0 = (1 + 2\lambda_2)s_j = |y_j| - \lambda_1, \tag{56}$$

which implies that all nonzero $s_j$ must satisfy $|y_j| = \lambda_1 = \max_j |y_j|$. Let $\mathcal{I}_{\max}$ be the index set where this holds:

$$\mathcal{I}_{\max} = \left\{ j \in \{pB, \ldots, pB + B - 1\} \,\middle|\, |y_j| = \max_{j'} |y_{j'}| \right\}, \quad \text{and let } m = |\mathcal{I}_{\max}|. \tag{57}$$

The two constraints require:

$$\sum_{j \in \mathcal{I}_{\max}} s_j = \tfrac{\sqrt{\pi}}{2} B, \tag{58}$$

$$\sum_{j \in \mathcal{I}_{\max}} s_j^2 = B. \tag{59}$$

To derive a lower bound on $m$, we apply the Cauchy–Schwarz inequality:

$$\left( \sum_{j \in \mathcal{I}_{\max}} s_j \right)^2 \leq m \cdot \sum_{j \in \mathcal{I}_{\max}} s_j^2. \tag{60}$$

Substituting from equation 58 and equation 59, we obtain:

$$\left( \tfrac{\sqrt{\pi}}{2} B \right)^2 \leq m \cdot B \quad \Rightarrow \quad \tfrac{\pi}{4} B^2 \leq mB \quad \Rightarrow \quad \tfrac{\pi}{4} B \leq m. \tag{61}$$

Thus, in order for the degenerate case $1 + 2\lambda_2 = 0$ to admit a feasible solution, at least $\tfrac{\pi}{4} \approx 78.5\%$ of the block entries (13 entries when $B = 16$) must have magnitudes exactly equal to $\max_j |y_j|$.

In practice, this is highly unlikely to occur since $y_j$ are continuous values, unless they are artificially set. Nevertheless, to safeguard against this rare but theoretically possible edge case, we introduce a small perturbation to the input:

$$\boldsymbol{y}^{(p)} \leftarrow \boldsymbol{y}^{(p)} + \delta\boldsymbol{\epsilon}, \quad \text{where} \quad \delta = 10^{-6}, \quad \boldsymbol{\epsilon} \sim \mathcal{N}(\boldsymbol{0}, \boldsymbol{I}), \tag{62}$$

which ensures that ties in magnitude are broken and uniqueness of the solution is preserved under typical 64-bit floating-point arithmetic.

Now, we consider the general case where $1 + 2\lambda_2 \neq 0$. In this case, the optimal magnitudes are given by:

$$s_j = \frac{1}{1 + 2\lambda_2} \cdot \mathrm{ReLU}\left(|y_j| - \lambda_1\right). \tag{63}$$

To enforce the constraints, we define the following two functions:

$$p_1(\lambda_1) := \sum_{j=pB}^{pB+B-1} \mathrm{ReLU}\left(|y_j| - \lambda_1\right), \tag{64}$$

$$p_2(\lambda_1) := \sum_{j=pB}^{pB+B-1} \mathrm{ReLU}\left(|y_j| - \lambda_1\right)^2. \tag{65}$$

Substituting $s_j$ into the constraint equations yields:

$$\sum_j s_j = \frac{1}{1 + 2\lambda_2} \cdot p_1(\lambda_1) = \tfrac{\sqrt{\pi}}{2} B, \tag{66}$$

$$\sum_j s_j^2 = \frac{1}{(1 + 2\lambda_2)^2} \cdot p_2(\lambda_1) = B. \tag{67}$$

From equation 66 and 67, we eliminate $\lambda_2$ to derive the identity:

$$\frac{p_1^2(\lambda_1)}{p_2(\lambda_1)} = \frac{\pi}{4}B. \tag{68}$$

This relation provides an equation of $\lambda_1$. Note that $\frac{p_1^2(\lambda_1)}{p_2(\lambda_1)}$ is a decreasing function with respect to $\lambda_1$ on the range $(-\infty, \max_j |y_j|)$. The value $\frac{p_1^2(\lambda_1)}{p_2(\lambda_1)}$ approaches $B$ as $\lambda_1 \to -\infty$, and approaches 0 as $\lambda_1 \to \max_j |y_j|$.

For efficient optimal value search, we define $\boldsymbol{w}$ to be the descending-sorted array of magnitudes within the block:

$$\boldsymbol{w} := \text{sort descending}\left(\{\, |y_{pB}|, \ldots, |y_{pB+B-1}| \,\}\right), \quad k = 0, \ldots, B-1. \tag{69}$$

Define the cumulative sums

$$S_{1,k} := \sum_{l=0}^{k} w_l, \quad S_{2,k} := \sum_{l=0}^{k} w_l^2. \tag{70}$$

Then, for $\lambda^{(k)} \in [w_{k+1}, w_k)$ (with $w_B := -\infty$ for convenience), we compute:

$$p_1\left(\lambda^{(k)}\right) = \sum_{l=0}^{k}\left(w_l - \lambda^{(k)}\right) = S_{1,k} - (k+1)\lambda^{(k)}, \tag{71}$$

$$p_2\left(\lambda^{(k)}\right) = \sum_{l=0}^{k}\left(w_l - \left(\lambda^{(k)}\right)^2\right) = S_{2,k} - 2\lambda^{(k)}S_{1,k} + (k+1)\left(\lambda^{(k)}\right)^2. \tag{72}$$

Substituting into the constraint $\frac{p_1^2(\lambda^{(k)})}{p_2(\lambda^{(k)})} = \frac{\pi}{4}B$ (letting $\gamma := \frac{\pi}{4}$), we obtain:

$$\frac{p_1^2(\lambda^{(k)})}{p_2(\lambda^{(k)})} = \gamma B, \tag{73}$$

$$p_1^2(\lambda^{(k)}) = \gamma B \cdot p_2(\lambda^{(k)}), \tag{74}$$

$$\left(S_{1,k} - (k+1)\lambda^{(k)}\right)^2 = \gamma B \cdot \left(S_{2,k} - 2\lambda^{(k)}S_{1,k} + (k+1)\left(\lambda^{(k)}\right)^2\right). \tag{75}$$

Rearranging yields the quadratic equation:

$$(k+1)(\gamma B - k - 1)\left(\lambda^{(k)}\right)^2 - 2S_{1,k}(\gamma B - k - 1)\lambda^{(k)} + \gamma B S_{2,k} - S_{1,k}^2 = 0. \tag{76}$$

To ensure real roots, we evaluate the discriminant:

$$\Delta = S_{1,k}^2(\gamma B - k - 1)^2 - (k+1)(\gamma B - k - 1)\left(\gamma B S_{2,k} - S_{1,k}^2\right) \tag{77}$$

$$= (\gamma B - k - 1)\left[S_{1,k}^2(\gamma B - k - 1) - (k+1)(\gamma B S_{2,k} - S_{1,k}^2)\right] \tag{78}$$

$$= (\gamma B - k - 1)\left[\gamma B S_{1,k}^2 - (k+1)\gamma B S_{2,k}\right] \tag{79}$$

$$= \gamma B(\gamma B - k - 1)\left(S_{1,k}^2 - (k+1)S_{2,k}\right) \tag{80}$$

$$= \gamma B(k + 1 - \gamma B)\left((k+1)S_{2,k} - S_{1,k}^2\right). \tag{81}$$

We clarify that the term $(k+1)S_{2,k} - S_{1,k}^2$ is nonnegative by the Cauchy–Schwarz inequality:

$$(S_{1,k})^2 = \left(\sum_{\ell=0}^{k} w_\ell\right)^2 \leq (k+1)\sum_{\ell=0}^{k} w_\ell^2 = (k+1)S_{2,k}. \tag{82}$$

Therefore, real-valued solutions exist only when $k + 1 \geq \gamma B$. Otherwise, the solution falls under the degenerate case previously handled in equation 61.

Thus, the solution for $\lambda^{(k)}$ is given by:

$$\lambda^{(k)} = \frac{S_{1,k} \pm \frac{\sqrt{\Delta}}{k+1-\gamma B}}{k+1}, \tag{83}$$

$$= \frac{S_{1,k} - \frac{\sqrt{\Delta}}{k+1-\gamma B}}{k+1} \quad \left( \text{since } \frac{S_{1,k}}{k+1} \geq w_k \right), \tag{84}$$

$$= \frac{S_{1,k}}{k+1} - \frac{\sqrt{\gamma B}}{k+1} \sqrt{\frac{(k+1)S_{2,k} - S_{1,k}^2}{k+1-\gamma B}}. \tag{85}$$

If $w_k > \lambda^{(k)} \geq w_{k+1}$, then this $\lambda^{(k)}$ is the solution $\lambda_1$ for equation 68. Once $\lambda_1$ is determined, the corresponding $\lambda_2$ is recovered from equation 66.

The final solution is given by:

$$s_j = \frac{1}{1 + 2\lambda_2} \text{ReLU}\left(|y_j| - \lambda_1\right) \tag{86}$$

$$= \frac{\sqrt{\pi B}}{2p_1(\lambda_1)} \text{ReLU}\left(|y_j| - \lambda_1\right) \quad \text{(by equation 66)}, \tag{87}$$

and

$$\dot{y}_j = \frac{\sqrt{\pi B}}{2p_1(\lambda_1)} \text{ReLU}\left(|y_j| - \lambda_1\right) e^{i\theta_j} \tag{88}$$

where $\theta_j$ is $\arg(y_j)$ if $y_j \neq 0$, and a random angle otherwise.

In practice, exact zero $y_j$ in 64-bit floating-point representations are extremely rare unless artificially introduced. Nonetheless, when this occurs, we perturb $y_j$ by adding a small white Gaussian noise $\delta\epsilon$, where $\delta = 10^{-6}$ and $\epsilon \sim \mathcal{CN}(0, 1)$.

### D.1 COMPUTATIONAL PERSPECTIVE

From a computational standpoint, the dominant cost arises from sorting the magnitudes $|y_j|$ within each block, which takes $\mathcal{O}(B \log B)$ time. Since there are $P$ blocks in total, the overall sorting cost is $\mathcal{O}(N \log B)$. All subsequent operations—such as computing prefix sums, evaluating the discriminant, and updating the filtered vector—can be performed in linear time per block. Therefore, the total runtime of the algorithm is $\mathcal{O}(N \log B)$.

Moreover, because each block is processed independently, the entire procedure is naturally parallelizable and well-suited for GPU acceleration.

## E RANGE OF THE MAGNITUDE SPECTRUM

In this subsection, we analyze the range of $|y_j|$. Since the constraints are permutation-invariant, we consider $|y_0|$ without loss of generality. The extreme cases occur when $|y_0|$ is the largest among the $B$ elements in the same block, and the remaining $(B-1)$ elements have equal magnitude. Let $a := |y_0|$ and $b := |y_1| = |y_2| = \cdots = |y_{B-1}|$. Under these assumptions, the following constraints hold (with $\gamma := \frac{\pi}{4}$):

$$a + (B-1)b = \sqrt{\gamma B}, \qquad a^2 + (B-1)b^2 = B. \tag{89}$$

Solving for $b$ in terms of $a$, we have:

$$b = \frac{\sqrt{\gamma B} - a}{B - 1}. \tag{90}$$

Substituting this into the second equation yields:

$$a^2 + (B-1)\left(\frac{\sqrt{\gamma B} - a}{B - 1}\right)^2 = B, \tag{91}$$

which simplifies to:

$$a^2 - 2\sqrt{\gamma}a + \gamma B - B + 1 = 0. \tag{92}$$

Solving this quadratic in $a$, we obtain:

$$a = \sqrt{\gamma} + \sqrt{(1-\gamma)(B-1)}, \quad b = \sqrt{\gamma} - \sqrt{\frac{1-\gamma}{B-1}}. \tag{93}$$

Thus, the extreme values that $|y_0|$ can attain under these constraints are:

$$|y_0|_{\min} = \sqrt{\gamma} - \sqrt{(1-\gamma)(B-1)}, \tag{94}$$

$$|y_0|_{\max} = \sqrt{\gamma} + \sqrt{(1-\gamma)(B-1)}. \tag{95}$$

For block size $B = 16$, we find $|y_0|_{\min} < 0$ and $|y_0|_{\max} \approx 2.68$. Therefore, $|y_j|$ lies in the range $[0.0, 2.68]$, which corresponds to approximately 99.92% coverage under the $\chi_2/\sqrt{2}$ distribution.

## F  RELATIONSHIP BETWEEN FLAT MAGNITUDE SPECTRUM AND ZERO AUTOCORRELATION

Let $\boldsymbol{x} \in \mathbb{R}^N$ and let $\hat{\boldsymbol{x}} = \boldsymbol{F}\boldsymbol{x}$ be the *unitary* DFT. Define the *circular* (period-$N$) autocorrelation

$$r_{\boldsymbol{x}}[\ell] = \frac{1}{N} \sum_{n=0}^{N-1} x_n \, x_{n-\ell \; (\mathrm{mod} \; N)}, \qquad \ell = 0, \ldots, N-1. \tag{96}$$

According to discrete Wiener–Khinchin relations (Khintchine, 1934), the periodogram $|\hat{x}_k|^2$ and the circular autocorrelation form an exact DFT pair:

$$|\hat{x}_k|^2 = \sum_{\ell=0}^{N-1} r_{\boldsymbol{x}}[\ell] \, e^{-2\pi i k\ell/N}, \quad k = 0, \ldots, N-1, \tag{97}$$

$$r_{\boldsymbol{x}}[\ell] = \frac{1}{N} \sum_{k=0}^{N-1} |\hat{x}_k|^2 \, e^{2\pi i k\ell/N}, \quad \ell = 0, \ldots, N-1. \tag{98}$$

Taking expectations of both sides yields the Wiener–Khinchin relations in expectation:

$$\mathbb{E}\big[|\hat{x}_k|^2\big] = \sum_{\ell=0}^{N-1} \mathbb{E}[r_{\boldsymbol{x}}[\ell]] \, e^{-2\pi i k\ell/N}, \quad k = 0, \ldots, N-1, \tag{99}$$

$$\mathbb{E}[r_{\boldsymbol{x}}[\ell]] = \frac{1}{N} \sum_{k=0}^{N-1} \mathbb{E}\big[|\hat{x}_k|^2\big] \, e^{2\pi i k\ell/N}, \quad \ell = 0, \ldots, N-1. \tag{100}$$

Let $c := \mathbb{E}[\, r_{\boldsymbol{x}}[0]\,] = \frac{1}{N} \, \mathbb{E}\big[\|\boldsymbol{x}\|_2^2\big]$. Then by equation 100,

$$\mathbb{E}[r_{\boldsymbol{x}}[\ell]] = c \, \delta_N[\ell] \quad \Longleftrightarrow \quad \mathbb{E}\big[|\hat{x}_k|^2\big] \equiv c, \tag{101}$$

where $\delta_N[\ell] = 1$ if $\ell \equiv 0 \pmod{N}$ and 0 otherwise. Hence a flat expected magnitude spectrum is *identical* to zero expected autocorrelation at all nonzero circular lags.

For a single finite realization, $|\hat{x}_k|^2$ fluctuates around its expectation, so perfect flatness is not observed. Nevertheless, promoting a well-spread (near-flat) magnitude distribution across frequencies reduces off-origin correlations via equation 98.

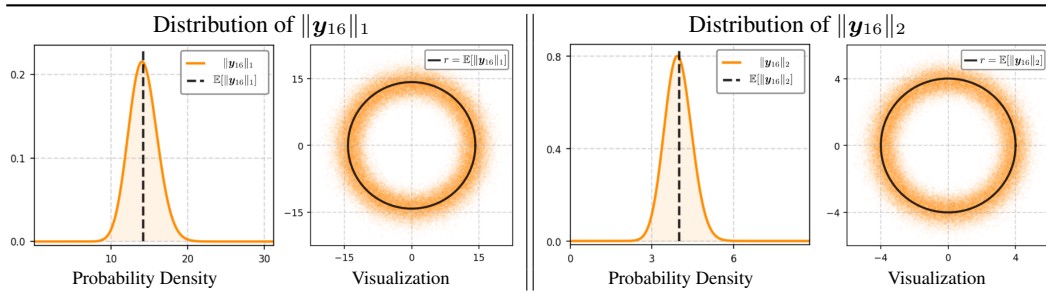

Figure 4: **Distributions of $\ell_1$ and $\ell_2$ norms of $\boldsymbol{y}_{16} \sim \mathcal{CN}(\mathbf{0}, \boldsymbol{I}_{16})$.** The radial plots visualize 100K samples, where radius indicates the norm and angle indicates $\arg(\mathbf{1}^\top \boldsymbol{y}_{16})$. In both cases, the norms are concentrated around their expected values.

## G  EFFECT OF BLOCK SIZE

In this section, we discuss the effect of the block size $B$ and present related experimental results.

Our constraints match the blockwise $\ell_1$ and $\ell_2$ norms to their theoretical expectations under the Gaussian distribution. As the block size $B$ decreases, the variance of these blockwise norms increases, and the feasible set moves farther away from the set of white Gaussian noise. Conversely, as $B$ increases, the blockwise norms of white Gaussian noise concentrate around their expectations, and the feasible set becomes closer to white Gaussian noise. At the same time, however, the feasible set becomes *less tight* for larger $B$, since the number of constraints decreases with the number of blocks.

This trade-off can also be seen from the allowable magnitude range derived in Section E. For example, the maximum allowed value of $|y_j|$ is 2.11 when $B = 8$, but increases to 84.74 when $B = 32,768$. This illustrates how the block size controls the tightness of the constraints around the notion of white noise: smaller blocks enforce tighter local control, while larger blocks permit much larger deviations in individual coefficients.

Based on this analysis, we set $B = 16$, which yields a feasible set that remains close to white Gaussian noise (the minimum cosine similarity with 1M samples from $\mathcal{N}(\mathbf{0}, \boldsymbol{I}_N)$ with $N = 65,536$ is 0.988), while still imposing sufficiently tight constraints of white Gaussian noise. The theoretical distributions of the blockwise $\ell_1$ and $\ell_2$ norms are shown in Figure 4.

| Block size $B$ | MPGR (baseline) | 8 | **16** | 32 | 64 | 32,768 |
|---|---|---|---|---|---|---|
| Aesthetic Score (target) | 7.1329 | 8.5883 | 8.9078 | **9.0833** | 8.1558 | 7.8834 |
| PickScore (held-out) | 0.2195 | 0.2177 | 0.2203 | 0.2193 | 0.2161 | 0.2110 |
| HPSv2 (held-out) | 0.2922 | 0.2940 | 0.2986 | 0.3111 | 0.2828 | 0.2623 |

Table 1: Effect of block size $B$ on FLUX-based reward-guided generation.

We further evaluate the effect of $B$ on FLUX (Labs, 2024) when optimizing the Aesthetic Score (Schuhmann et al., 2022). Table 1 reports the target reward (Aesthetic Score) and held-out rewards (PickScore and HPSv2) for different block sizes. For all $B$, our method improves the Aesthetic Score over MPGR (Hwang et al., 2025). However, the held-out rewards degrade as $B$ becomes very large, reflecting the looser nature of the constraints. In contrast, the performance for $B = 8$, 16, and 32 is comparable in both target and held-out metrics, indicating that the method is not overly sensitive around our chosen value $B = 16$.

## H  FEWER ITERATIONS TO REACH COMPARABLE REWARD

Our constrained optimization framework maximizes the target reward more efficiently than regularization-based baselines in the standard test-time latent optimization setting. In practice, this means that it can reach a comparable reward level with substantially fewer gradient-ascent iterations, and hence significantly shorter wall-clock time. Here, we present experimental results on FLUX for maximizing the Aesthetic Score.

| Method | No Opt. 0 iters. | ReNO 200 iters. | PRNO 200 iters. | MPGR 200 iters. | **Ours** *60 iters.* | Ours *200 iters.* |
|---|---|---|---|---|---|---|
| Aesthetic Score (target) | 5.9932 | 7.0580 | 7.0244 | 7.1329 | 7.1195 | 8.9078 |
| PickScore (held-out) | 0.2192 | 0.2193 | 0.2183 | 0.2195 | 0.2202 | 0.2203 |

Table 2: Comparison of FLUX-based reward-guided generation under different iteration counts.

As shown in Table 2, our method reaches an Aesthetic Score comparable to MPGR with only 60 iterations (7.1195 vs. 7.1329), whereas all baselines are evaluated at 200 iterations. The held-out reward (PickScore) remains essentially unchanged across methods. In our setup, 200 iterations correspond to about 4 minutes of computation, while 60 iterations take only about 1 minute 12 seconds. This demonstrates that our constrained optimization achieves similar or better reward levels with roughly 30% of the iterations and time required by regularization-based approaches, making test-time latent optimization substantially more practical.

## I  MAGNITUDE DISTRIBUTIONS ACROSS OPTIMIZATION METHODS

To complement the quantitative statistics, we further examine how different latent optimization methods affect the empirical distributions of spectral magnitudes. For a fixed latent sample, we compute the elementwise magnitudes $|y_i|$ as well as the blockwise norms $\|\boldsymbol{y}^{(p)}\|_1$ and $\|\boldsymbol{y}^{(p)}\|_2$ of the complex-valued spectrum. For each method, we plot these quantities for both a white Gaussian noise and the optimized latent vector. For our method, since the blockwise norms are fixed, we instead show the empirical probability density function of $|y_i|$.

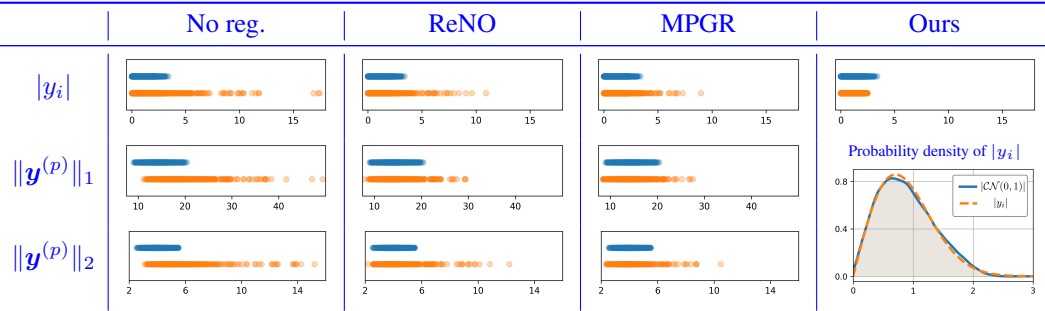

Figure 5: Empirical distributions of spectral magnitudes for each optimization method. ■ indicates values obtained from a white Gaussian noise, and ■ indicates values from the optimized latent vector. For each frequency or block $p$, the corresponding magnitude or norm is plotted at its location on the horizontal axis. The rightmost panel shows the empirical and theoretical probability density of $|y_i|$.

Compared to the unregularized optimization, ReNO and MPGR reduce the deviation of the empirical magnitudes from those of the white Gaussian noise. However, the $\ell_1$ distributions of MPGR still exhibit noticeable deviations, indicating that soft penalties on $\|\boldsymbol{y}^{(p)}\|_1$ do not fully prevent drift in blockwise magnitudes. Moreover, the regularization-based methods produce individual frequencies whose magnitudes are substantially larger than expected under the white Gaussian noise. In contrast, our constrained method keeps elementwise magnitudes around the Gaussian distribution, and the upper bound is set (Appendix E). This behavior is consistent with the notion of white noise discussed in Section 4.5, where no single frequency component is disproportionately strong.

## J  EXPERIMENTAL RESULTS WITH SDXL-TURBO

We report quantitative and qualitative results with SDXL-Turbo in Figures 6 and 7, respectively. Across all reward models, our method consistently achieves better trade-offs than the baselines while not significantly losing held-out rewards compared to the unoptimized outputs (No Opt.), thereby improving reward alignment without compromising realism or image quality. By contrast, baselines

across different learning rates and regularization schemes fail to achieve our trade-offs. We reason this limitation arises from soft regularization, which provides no guarantee of white Gaussian noise constraints even with sophisticated loss formulations.

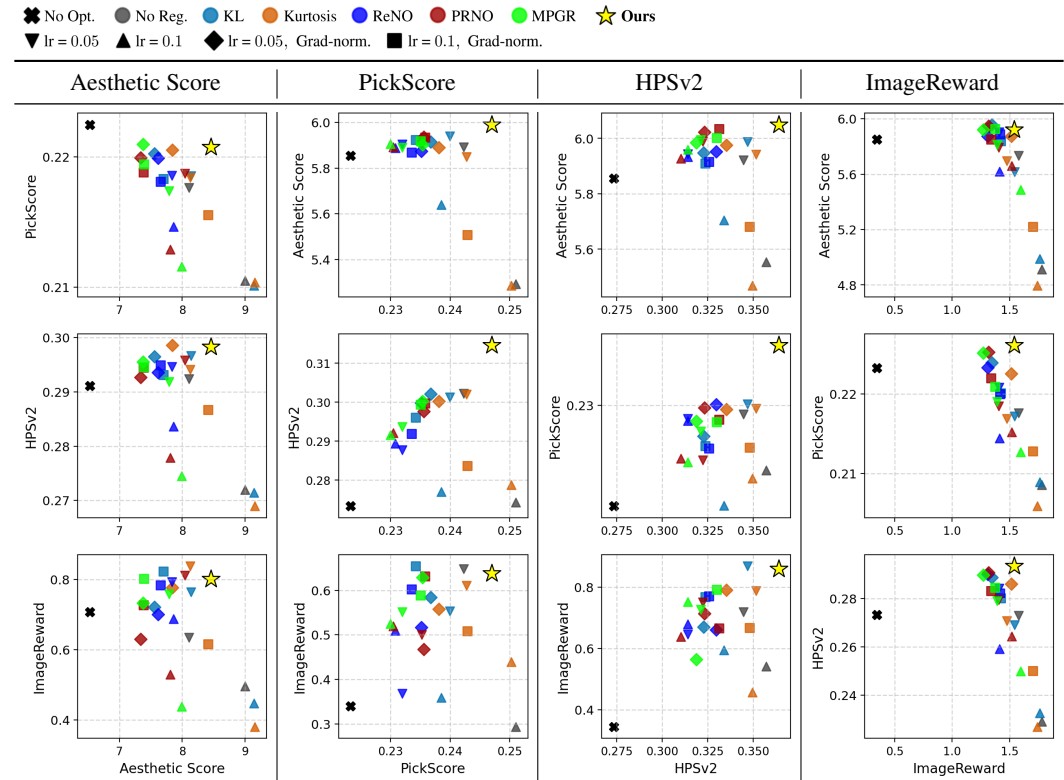

Figure 6: **Quantitative Results with SDXL-Turbo.** Each column corresponds to the same given reward (x-axis), and different held-out rewards (y-axis). Each point denotes the score after 50 iterations, with higher positions and more rightward placement indicating better trade-offs. For baselines, multiple points are plotted across learning rates and regularization schemes.

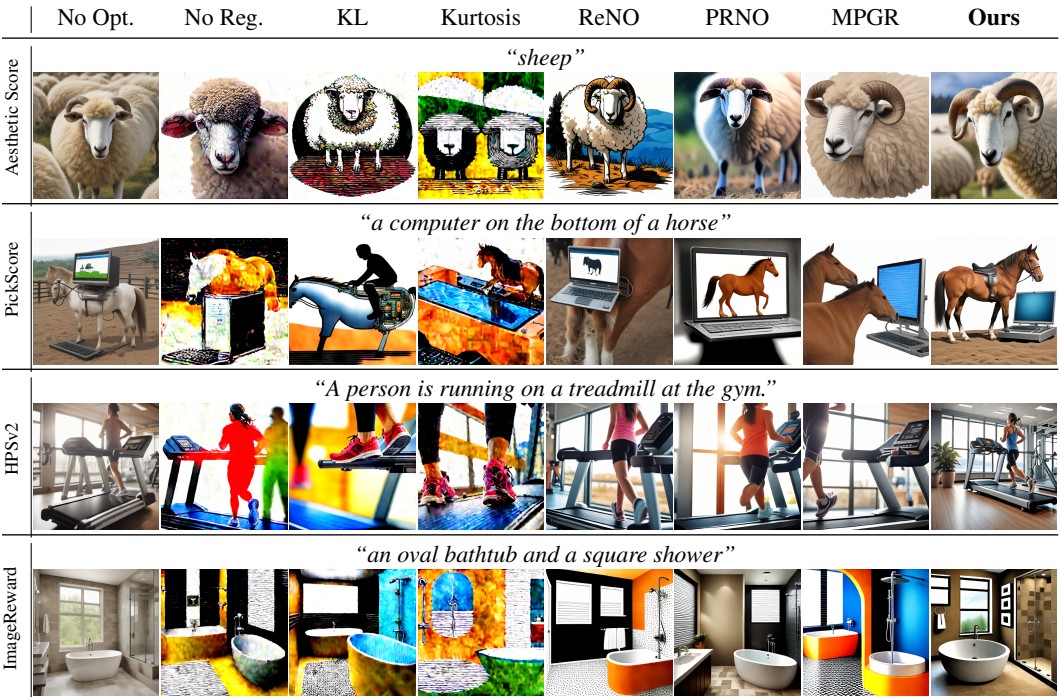

Figure 7: **Qualitative results with SDXL-Turbo.** Columns denote optimization method; rows correspond to the given reward, with the prompt shown above each row. Our constrained optimization preserves realism and prompt fidelity while attaining higher target scores and strong held-out quality.

## K LLM USAGE

We used a large language model to identify typos and refine phrasing; all logic and methods were designed by us.

