# OpenReview forum: "White Gaussian Noise Constraints for Reward-Guided Generation"
_ICLR.cc/2026/Conference — Submitted to ICLR 2026_

### Official Review · Reviewer_ERkz · 2025-10-27

**Soundness:** 3
**Presentation:** 1
**Contribution:** 3
**Rating:** 6
**Confidence:** 4

**Summary:**

The paper presents a constrained latent optimization framework for diffusion model reward maximization. Latents are projected to the intersection of L1 and L2 norm spheres that match the first and second moments of a gaussian distribution. They present empirical gains with respect to penalized and unconstrained approaches for aesthetic/preference rewards.

**Strengths:**

I think that both the use of constrained optimization tools for latent diffusion optimization is a valuable contribution that could inspire future work in this direction. I believe that the empirical evidence strongly supports the advantages of their methods with respect to baselines on terms of performance, and that their approach is computationally efficient, and thus widely applicable.

**Weaknesses:**

I think that the theoretical and algorithmic contributions of this paper need to be properly outlined and references to prior work have to be added.

The "construction of a compact spectral domain", i.e. using the one-sided DFT, is customary in spectral analysis of real valued signals. That the DFT is hermitian (Lemma 1) and the Fourier transform of a gaussian is a gaussian (Theorem 1) are standard results. I think it would be useful to point this out and at least include a reference to a standard signal processing textbook.

In the same vein, algorithms projecting into the intersection of the L1 and L2 spheres (the algorithm presented in section 4.3)  have been studied before, see e.g. [1]. No reference to prior works are included here.

I am not saying the use of these tools is not appropriate or that their application in this problem is not novel, just that their relation to prior work is missing altogether, and their presentation might mislead the reader into thinking these are contributions of this work.

[1] Liu, H., Wang, H., & Song, M. (2019). A unified approach for projections onto the intersection of $\ell_1$ and $\ell_2$ balls or spheres.

**Questions:**

Can you expand a bit more on why constraining the l1 and l2 norm to exactly match their expected values under gaussianity is beneficial?

Can you also discuss the use of blocks instead of constraints on the whole vector, and the impact of block size choice?

Can you plot the resulting L1 and L2 distributions of baseline latent optimization methods?

---

> ### Author Response · Authors · 2025-11-26
>
> We sincerely thank the reviewer for taking the time to review our paper and for highlighting both the conceptual value and empirical strength of our approach.
>
> ---
>
> ### [W1] Theoretical / algorithmic contributions and missing references
>
> We thank the reviewer for pointing out the need to more clearly distinguish our contributions from standard background material and to add appropriate references. To avoid any misunderstanding, we have revised the paper as follows:
>
> - We added a textbook reference [1] for the Hermitian symmetry of the DFT coefficients (Lemma 1, **L597**).
> - We clarified that the existence of a closed-form projection onto the intersection of $\ell_1$ and $\ell_2$ spheres is known and cited [2] in Section 4.3 (**L265-269**).
>
> As the reviewer notes, our novelty does not lie in restating these standard facts. Instead, it lies in **formulating constraints and a compact spectral mapping** that (i) exploit Hermitian symmetry to obtain a compact, one-sided representation and independent blocks, and (ii) allow the closed-form $\ell_1$-$\ell_2$ projection to be applied blockwise in this domain. This design enables projected gradient ascent with negligible overhead (0.04% of runtime for the projection, **L288**) while mathematically guaranteeing white Gaussian noise characteristics, in particular that no single frequency can disproportionately dominate (**L350-356**, **L984**), and this in turn leads to consistently stronger results on test-time latent optimization tasks.
>
> - [1] Digital signal processing: principles, algorithms, and applications, 4th Edition (Proakis, 2007)
> - [2] A unified approach for projections onto the intersection of $\ell_1$ and $\ell_2$ balls or spheres. (Liu et al., 2019)
>
> ---
>
> ### [Q1, Q2] Why constrain the $\ell_1$ and $\ell_2$ norms to their Gaussian expectations, and why use blockwise constraints with this particular block size $B$?
>
> We clarified this design in Section 4.5 (**L350-356**) and Appendix E, and in the revision we added further justification in Section 4.2 (**L216-221**) and Appendix G. Our constraints **follow the Gaussian statistics** of blockwise $\ell_1$ and $\ell_2$ norms, but by enforcing them as **hard constraints** we additionally prevent any single frequency from acquiring a **disproportionately large magnitude**, which should be avoided under the notion of *white noise* (**L350-356**).
>
> As shown in Appendix E, under our blockwise constraints the **theoretical upper bound** on each spectral magnitude depends only on the block size $B$.
>  - When $B$ is very **large**, this bound becomes extremely loose and still allows a few frequencies to dominate the spectrum. (**L216-221**)
>  - When $B$ is very **small**, the higher variance of Gaussian block statistics means that enforcing exact equality makes the feasible set deviate far from white Gaussian noise (Appendix G, **L1044-1045**).
>
> The block size $B = 16$ in our experiments balances between these two extremes:
>  - The constraints impose a **moderate spectral bound**: the theoretical maximum of $|y_j|$ is $2.68$ (covering 99.92\% of the theoretical distribution, **L984**), guaranteeing that no single frequency can dominate.
>  - The feasible set remains **close to white Gaussian noise**: the minimum cosine similarity between $1$M samples of white Gaussian noise and their projections onto $\mathcal{G}_{\mathbb{R}}$ is $0.988$ for $N = 65{,}536$ (**L290-292**).
>
> ---
>
> ### [Q3] Can you plot the resulting $\ell_1$ and $\ell_2$ distributions of baseline latent optimization methods?
>
> We added **Appendix I**, where we compare the **empirical distributions of compact spectral magnitudes** across optimization methods. For each method, we plot:
>
> - elementwise magnitudes $|y_i|$, and
> - blockwise norms $\|y^{(p)}\|_1$ and $\|y^{(p)}\|_2$ in the compact spectral domain,
>
> for both a white Gaussian noise and the optimized latent vector. The results are summarized in **Figure 5**.
>
> Compared to unregularized optimization, ReNO and MPGR partially reduce the deviation from the white Gaussian noise, but still produce blocks and individual frequencies with much larger magnitudes. In contrast, our constrained method keeps the empirical distributions of $|y_i|$ consistent with the theoretical bounds from Appendix E and the white noise perspective in Section 4.5.
>
> ---
>
> With the clarifications and additions, we believe the revised manuscript now clearly separates standard background from our original contributions, namely a white Gaussian noise feasible set in a compact spectral domain, a mathematically justified blockwise projection with negligible overhead, and consistent gains over strong baselines across multiple rewards. We hope this helps convey the significance and robustness of our contribution within the broader landscape of test-time latent optimization methods.

---

### Official Review · Reviewer_738g · 2025-10-27

**Soundness:** 3
**Presentation:** 2
**Contribution:** 3
**Rating:** 4
**Confidence:** 3

**Summary:**

The paper introduces a constrained optimization framework for reward-guided generation that explicitly enforces Gaussianity in the latent space. Instead of using soft regularization (as in previous methods like PRNO or MPGR), the authors propose a closed-form projection that ensures the latent vector remains within a feasible set representing white Gaussian noise.

**Strengths:**

- The idea of replacing regularization with explicit Gaussian constraints and deriving a closed-form projection is original and mathematically elegant.
- The use of a compact spectral domain mapping (bijective and preserving Gaussianity) is both theoretically sound and practically efficient.

**Weaknesses:**

The experimental results are limited.
- The number of testing prompts is limited to just 60 prompts, which is a very small set of samples. For evaluation, a set of at least 1000 prompts is needed.
- The testing dataset's domain is limited to animals.
- The testing problem is limited to text-to-image generation. I'm curious about the model performance when being tested with different problems.

**Questions:**

Please see the above weaknesses.

---

> ### Author Response · Authors · 2025-11-26
>
> We thank the reviewer for their comments on the scope and design of our experiments. Below, we address each concern in turn.
>
> ---
>
> ### [W2] On the claim that the testing dataset’s domain is limited to animals
>
> Our experiments are **not** limited to animal prompts. Only **two out of eight** qualitative examples in the paper (**Figures 3** and **6**) correspond to animal prompts, and our quantitative experiments are conducted on the **T2I-CompBench++** dataset (**L417**), which contains a **diverse set of compositional text prompts**, not an animal-specific dataset.
>
> ---
>
> ### [W1, W3] Concerns about the experimental setup
>
> We would like to emphasize that our experimental setup **directly follows** the latest baseline **MPGR (Hwang et al., NeurIPS 2025)**, including the task (one-step text-to-image generation) and the datasets. On top of their setup, we **extended** the evaluation by:
>
> - adding additional reward models (**HPSv2** and **ImageReward**), and
> - exploring more regularization settings (more learning rates and regularization schemes) for the baselines,
>
> in order to provide a more comprehensive and fair assessment of our method.
>
> In addition to the 60-prompt experiment reported in the main paper, we now present a new experiment on **1000 prompts** from **T2I-CompBench++**, using **FLUX** with **PickScore** as the target reward:
>
> | 1000 prompts | No Opt. | PRNO   | MPGR   | Ours   |
> |---|---:|---:|---:|---:|
> | PickScore (target) ↑ | 0.2270 | 0.2444 | 0.2438 | **0.2660** |
> | Aesthetic Score (held-out) ↑ | 5.7060 | 5.6856 | 5.7056 | 5.6161 |
> | HPSv2 (held-out) ↑ | 0.2856  | 0.3023 | 0.3031 | 0.3152 |
>
> Relative to “No Opt.”, PRNO and MPGR improve PickScore by about **0.017–0.018**, whereas our method improves it by **0.039**, more than **twice** the gain achieved by the baselines. At the same time, Aesthetic Score and HPSv2 remain comparable or slightly better for our method.
>
> This confirms that the tendency observed in the 60-prompt setting persists on a substantially larger evaluation set: our constrained optimization consistently yields a much stronger improvement in the **target reward** while maintaining competitive held-out rewards.
>
> ---
>
> Overall, we hope that these clarifications, together with the expanded experiments building on the NeurIPS-accepted MPGR setup, will alleviate concerns regarding the scope and adequacy of our experimental evaluation.

---

### Official Review · Reviewer_fY13 · 2025-10-31

**Soundness:** 2
**Presentation:** 2
**Contribution:** 2
**Rating:** 4
**Confidence:** 3

**Summary:**

This paper introduces a novel inference-time input noise optimization technique for reward-guided generation based on projected gradient ascent (with closed-form projection) on a feasible set, which characterizes *Gaussianity* via block-wise $l_1$ and $l_{2}$ norm constraints of half-spectrum representation. The proposed feasible set constraints are presented to enforce the spatial and spectral characteristics of white Gaussian noise. Experiments show that the suggested method substantially outperforms similar baselines on *one-step* text-to-image human preference reward-guided generation.

**Strengths:**

1. Reformulation of test-time reward-guided optimization of the input noise via projected gradient ascent is novel;
2. The motivation to use the first two moments of magnitude of half-spectrum components as characterizations of *Gaussianity* is supported by strong empirical results compared to the chosen baselines;
3. Overall, the empirical performance of the method according to the quantitative evaluation on human-preference benchmarks is superior to the other input noise optimization methods.

**Weaknesses:**

1. Despite utilizing the closed-form projection, the approach relies on many gradient steps per prompt. This results in a highly impractical method that performs hundreds of backward passes through both the reward model and the diffusion model to perform just *one*  reward-guided inference with *one-step* model. If the method could generalize on different prompts, it would potentially improve its applicability. In its current form, I would rather treat the method as a demonstration of the potential in optimizing the latents, than a practical option.
2. The paper repeatedly frames the method as *preserving Gaussianity*, yet the feasible set enforces just the equalities of block-wise $l_{1}$ and $l_{2}$ norms in half-spectrum. Performing projection on this feasible set is not equivalent to maintaining the original Gaussian measure. Conversely, the resulting distribution will be concentrated on a manifold and will not have Lebesgue density. Moreover, this projection does not even guarantee that the distribution after projection will be similar to the Gaussian distribution projected on the same feasible set (e.g. projection of the Gaussian distribution on an $l_2$ sphere is uniform, while here this is an almost arbitrary distribution on the feasible set). The terminology and positioning seem misleading and could be read as stronger than what the constraints actually ensure.
3. Some parts of the manuscript seem to have either too little or too much description of the underlying observations. Sections 4.1 – 4.2, for example, revisit such classic results as DFT symmetry or Gaussian concentration (Figure 1) in detail. At the same time, the intuition behind the main projection algorithm is almost non-explained and is largely deferred to the Appendix, making the paper's main result less comprehensible.

**Questions:**

1. Could you please tell, how sensitive are the results of the method to the block size $B$?
2. Figure 2 claims MPGR requires a *slow gradient-based projection*, implying a large complexity gap between methods and a 4000$\times$ runtime difference. My understanding is that MPGR jointly optimizes reward with both spectral regularization and moment matching in the spatial domain, resulting in almost the same complexity as the proposed method. Could you please provide details for the setting used in Fig. 2, and what *projection* you attribute to MPGR in that comparison?
3. The method differs from the prior works in two ways: it introduces *hard* constraints and combines $l_{1}$ with $l_{2}$. Could the authors explain which of the two components is responsible for the gains compared to the prior methods: the combination of $l_{1}$ and $l_{2}$, or the explicit projection onto the feasible set? What would happen if we optimise soft constraints $L_{\text{norm}}$ and $L_{\text{power}}$ jointly with gradient ascent without any projections?

---

> ### Author Response · Authors · 2025-11-26
>
> We sincerely appreciate your careful reading and incisive comments, which led us to refine our terminology around white Gaussian noise characteristics, improve the structure of Section 4, and more clearly position our method as a constrained approach with a closed-form projection that offers both strong reward optimization and practical computational cost.
>
>
> Below, we respond to each weakness and question in turn.
>
> ---
>
> ### [W1] Practicality and computational cost of projected gradient ascent
>
> We respectfully disagree with the assessment that our method is “highly impractical” in the context of **test-time latent optimization**. In this setting, performing on the order of a few hundred backward passes per prompt is still vastly cheaper than retraining or fine-tuning the generative model itself, and can be carried out with relatively modest computational resources. This is precisely why prior works such as ReNO, PRNO, and MPGR adopt **per-prompt, test-time optimization of a latent vector** via gradient ascent and position it as a practical option for small- to medium-scale runs where retraining is infeasible or unjustified.
>
> Within this regime, **our objective is in fact to significantly improve practical usability**. For a given number of backward passes, our method attains substantially higher rewards (as shown in the **Section 5**), and conversely, to reach a given reward level it requires substantially fewer iterations and therefore less wall-clock time than regularization-based baselines. For example, on FLUX we reduce the runtime from about **4 minutes to roughly 1 minute 12 seconds per prompt** (≈70% reduction) while using the same type of backward passes as prior methods. (The projection step costs only 0.47 ms per iteration (**Figure 1**), making the per-iteration runtime practically identical.)
>
> With **60 iterations** (i.e., **30%** of the 200 iterations used by the baselines), our method already matches or slightly exceeds their performance:
>
> | Metric | No Opt. | ReNO (200 iters) | PRNO (200 iters) | MPGR (200 iters) | Ours (60 iters) | Ours (200 iters) |
> |---|---:|---:|---:|---:|---:|---:|
> | Aesthetic Score (target) ↑ | 5.9932 | 7.0580 | 7.0244 | 7.1329 | 7.1195 | 8.9078 |
> | PickScore (held-out) ↑ | 0.2192 | 0.2193 | 0.2183 | 0.2195 | 0.2202 | 0.2203 |
>
> We have added this discussion to **L466** and **Appendix H** to clarify that, rather than being less practical, our method *improves* the practicality of test-time latent optimization.
>
> ---
>
> ### [W2] Clarifying “Gaussianity” and the scope of our guarantees
>
> We appreciate the comment on terminology. While we adapted the term “Gaussianity” from previous work MPGR, it is true that the term may sound overclaiming, as the properties we focus on, namely **white Gaussian noise characteristics**, do not fully cover all properties of Gaussian distribution. To avoid confusion, we **removed all occurrences** of the term “Gaussianity” and instead used “white Gaussian noise characteristics” throughout the revision. Please check out our revised version.
>
> Our focus is to regularize a latent vector to *preserve white Gaussian noise characteristics* (both **Gaussian statistics** and **low spatial correlation**, **L46**), whose importance for the proper operation of generative models has been emphasized  in previous works: PRNO and MPGR (**L47-49**). For the low spatial correlation, we design our constraints in the (compact) spectral domain, and we use Gaussian statistics ($\ell_1$ and $\ell_2$ norms). Please note that these characteristics are not closely related to Lebesgue measure or distributional properties in $N$-dimensional space.
>
> ---
>
> ### [W3] Balance between background material and intuition
>
> We thank the reviewer for the feedback on paper structure. We have made the following changes in revision:
>
> - **Clarified the motivation at the start of Section 4.**
>   We introduce $L_{\text{power}}$ (**L150–159**) and use it to motivate:
>   - why we move to the **compact spectral domain** (**L161–168**), and
>   - why **blockwise** (as opposed to purely global) constraints are natural (**L169–170**).
>
> - **Moved basic background to the appendix.**
>   Classical preliminaries such as DFT symmetry and Gaussian concentration (previous **Lemma 1** and **Figure 1**) are now moved to the appendix (**Appendix Lemma 1** and **Figure 4**), so the main text is less burdened with standard facts.
>
> - **Added explicit intuition for the projection design.**
>   Sections 4.1–4.2 now:
>   - explain the role of the compact spectral mapping (**L181–184**),
>   - describe the intuition behind blockwise $\ell_1/\ell_2$ constraints (**L216–221**), and
>   - outline the main steps of the blockwise projection (**L265–269**) before referring to the appendix for details.
>
> We believe these changes make the core idea and algorithm significantly easier to follow, while keeping necessary background available in the appendix rather than in the main flow. Please check out our revised version.
>
> ---

---

> ### Author Response · Authors · 2025-11-26
>
> ---
>
> ### [Q1] Sensitivity to block size $B$
>
> We have added an ablation over the block size $B$ on FLUX with Aesthetic Score as the target reward (Appendix G):
>
> | Block size $B$ | MPGR (baseline) | 8 | 16 | 32 | 64 | 32,768 |
> |---|---:|---:|---:|---:|---:|---:|
> | Aesthetic (target) ↑ | 7.1329 | 8.5883 | 8.9078 | 9.0833 | 8.1558 | 7.8834 |
> | PickScore (held-out) ↑ | 0.2195 | 0.2177 | 0.2203 | 0.2193 | 0.2161 | 0.2110 |
> | HPSv2 (held-out) ↑ | 0.2922 | 0.2940 | 0.2986 | 0.3111 | 0.2828 | 0.2623 |
>
> All choices of $B$ improve the **target reward** over MPGR. For $B = 8, 16, 32$, the **held-out metrics remain very similar** and in fact slightly improve in some cases (e.g., HPSv2 for $B=32$). Only very large blocks ($B = 64$ and especially $B = 32{,}768$) start to degrade held-out scores, reflecting that the constraints become too coarse.
>
> This suggests that our method is **not highly sensitive** to the block size around our default choice $B = 16$, and that a broad range of moderate block sizes works well in practice.
>
> ---
>
> ### [Q2] Interpretation of Figure 2 and “4000×” projection cost for MPGR
>
> Thank you for pointing out that this distinction was not sufficiently clear.
>
> First, please note that the experiment shown in Figure 2 (**now Figure 1**) is **not** about reward maximization, but a toy setup designed solely to demonstrate the efficiency of our **closed-form projection**. In this figure, “Ours” denotes applying **a single closed-form projection** from the initial latent using our formulation. In contrast, the MPGR baseline (Hwang et al., 2025), which does **not** admit a closed-form projection, must **instead approximate the projection via gradient descent** on a soft regularizer, making its “projection” roughly 4000× slower than ours.
>
> Concretely, the gradient-based projection solves
> $$
>   \min_{\mathbf{x}} ||\mathbf{x} - \mathbf{x}_0||_2^2 + \lambda \mathcal{L}(\mathbf{x}).
> $$
> We optimize this objective using Adam optimizer with $\lambda = 5.0$, learning rate $0.1$, and 400 iterations (chosen as the minimal setting for which the FLUX outputs become realistic). Each iteration requires a full forward–backward pass, so this *gradient-based projection* is orders of magnitude more expensive than our *closed-form projection*, which only requires one pass.
>
> Please note that, in our **reward maximization** experiments, we do **not** perform projection for the MPGR baseline (Hwang et al., 2025), and we use the **same number of iterations** for both our method and MPGR, ensuring that all reported comparisons are fair.
>
> ---
>
> ### [Q3] Hard constraints vs. soft regularization
>
> We believe that the introduction of **hard constraints** is the main factor behind the performance gains. Note that MPGR already includes an $\ell_2$-type moment term together with a spectral $\ell_1$ term and is optimized as a **soft penalty** jointly with the reward. This setup essentially corresponds to optimizing soft constraints $\mathcal{L}_{\text{norm}}$ and $\mathcal{L}_{\text{power}}$ jointly with gradient ascent, without any projections.
>
> In our experiments, soft regularization methods, including MPGR, show only marginal differences among themselves and relatively limited improvement compared to our constrained formulation. As shown in Figure 2 (leftmost three plots), the points corresponding to different regularization methods (colors) occupy a similar region, they raise the Aesthetic Score from about 6.0 to at most 7.5 across learning rates and regularization schemes. In contrast, our hard-constrained formulation reaches around 9.0 while maintaining comparable held-out rewards. (**L462**)
>
> ---

---

> > ### Comment · Reviewer_fY13 · 2025-11-27
> >
> > Thanks for answering my questions and providing new experiments. I think that rewriting the paper in terms of "white Gaussian noise characteristics" is beneficial for positioning. I also appreciate ablation of the block size and the revised structure of the paper. My minor concern with regard to the projection toy experiment is that the significantly higher cost of performing optimization-based projection does not imply higher cost in the main task of reward optimization, thus may confuse the reader with the $4000\times$ comparison. I would suggest to only keep the qualitative analysis here, which is beneficial.
> >
> > My main concern remains the same. Despite the authors claim their method practical, it is only practical when compared to the other test-time noise optimization methods, which are all incredibly costly. Performing 60 to 200 backward passes through the model for **one** forward pass with the **one-step** model seem to be very restrictive for practical applications. If the method somehow manages to significantly reduce the order of total steps or manages to adapt between different prompts, then the optimization cost will be justified and will raise much less questions about practicality. The other possible direction is generalizing the method for multi-step models. Achieving the comparable number of optimization steps for them would also be much more justified due to the lower order of discrepancy between the original and the reward-tuned performance.
> >
> > I decided to raise my score to 6 due to my main concerns being more about the overall approach then the particular authors' method. However, I think that in its current form the paper is borderline, and the other reviewers' thoughts about the method's practicality would be highly beneficial for making the decision.

---

> > > ### Author Response · Authors · 2025-12-02
> > >
> > > ### **We sincerely appreciate your decision to raise the score to 6.**
> > >
> > > We would like to further clarify your remaining concerns as follows.
> > >
> > > ---
> > >
> > > Regarding the minor concern that the Figure 1 may be confusing, we have **removed the runtime statistics** from the figure and **removed the “4000×” mention**. We expect this modification to remove any confusion between the projection toy experiment and the main reward-optimization cost.
> > >
> > > ---
> > >
> > > To address your main concern about practicality, we consider it from two perspectives:
> > >
> > > ## 1. Methods applicable to general prompts: fine-tuning is much more costly, and orthogonal to test-time optimization
> > >
> > > Methods that make a model better for a **wide variety of prompts** typically rely on fine-tuning, for example RL-based approaches such as Flow-GRPO (Liu et al., NeurIPS 2025). Fine-tuning requires very long durations: Flow-GRPO operates on the order of **thousands of GPU hours** to fine-tune Stable Diffusion 3.5 Medium (2.5B parameters). This cost is paid **once per model and reward**, but it is extremely high compared to our **minutes-level** per-prompt optimization on FLUX (12B parameters).
> > >
> > > We also emphasize that **these two directions are orthogonal**. Even if a model has already been fine-tuned using Flow-GRPO or a similar method to improve average behavior over prompts, our **test-time optimization** can still be applied on top to further optimize a *single* important prompt. In other words, model-level alignment and per-prompt optimization can be used together rather than as alternatives. Therefore, we believe that progress in test-time optimization is as valuable as advances in the fine-tuning domain.
> > >
> > > ## 2. Our runtime and NFE are in line with existing test-time optimization methods
> > >
> > > We would like to clarify that, beyond the baselines presented in our paper, there is a large body of **test- (or inference-) time optimization** work that invests **minutes per prompt**. We summarize a few representative examples, all of which optimize **per prompt**, not across prompts:
> > >
> > > - *FreeDoM: Training-Free Energy-Guided Conditional Diffusion Model* (Yu et al., ICCV 2023)
> > >   **84 seconds** per image.
> > >
> > > - *D-Flow: Differentiating through Flows for Controlled Generation* (Ben-Hamu et al., ICML 2024)
> > >   **2.5–15.5 minutes** per image.
> > >
> > > - *Test-Time Alignment of Diffusion Models Without Reward Over-Optimization* (Kim et al., ICLR 2025 Spotlight)
> > >   **400 forward–backward passes** per prompt.
> > >
> > > - *Inference-Time Scaling for Flow Models via Stochastic Generation and Rollover Budget Forcing* (Kim et al., NeurIPS 2025)
> > >   **10 minutes 35 seconds** per prompt.
> > >
> > > - *Ψ-Sampler: Initial Particle Sampling for SMC-Based Inference-Time Reward Alignment in Score Models* (Yoon et al., NeurIPS 2025 Spotlight)
> > >   **500 forward–backward passes**, roughly over **10 minutes** per prompt.
> > >
> > > All of these methods are used in scenarios where users are willing to spend up to a few minutes of compute to obtain a single high-quality, reward-aligned output. We also believe that test-time optimization is particularly important when prompts are **few but important** (e.g., advertising, key illustrations, concept art). In a similar spirit, modern large-scale services (e.g., **Gemini 3** or **ChatGPT 5.1** with “thinking” modes) already use **inference-time scaling** to improve response quality per prompt, sometimes **investing minutes** per query.
> > >
> > > In this context, our experimental setup of **200 forward–backward passes and about 4 minutes** per prompt (or **1 minute 12 seconds** with comparable reward to baselines) lies well within the established range for test-time optimization. Thus, for the cases where such test- (or inference-) time optimization is considered, our method is competitive in terms of both **NFE** and **wall-clock time**.
> > >
> > > Taken together, we believe this clarifies that:
> > >
> > > - methods that improve behavior for **general prompts** via fine-tuning operate at a **much higher training cost** and can still **be complemented** by our test-time optimization, and
> > > - within the **per-prompt test-time optimization regime**, our runtime and NFE are in a **typical and acceptable range**, while providing stronger reward maximization and stability than prior noise-optimization baselines.
> > >
> > > We hope this additional context helps address your concern about practicality.
> > >
> > > ---
> > >
> > > We developed a novel framework of constrained optimization and proposed a concrete formulation that leverages properties of the Gaussian distribution to define a *white Gaussian noise feasible set* with a closed-form projection. Using this formulation, we demonstrated consistently strong empirical results in our experiments. It would be unfortunate if such an effective and conceptually clean formulation were not reported in the scholarly literature due to a runtime and NFE that are already typical for test-time optimization methods.

---

### Official Review · Reviewer_7U1f · 2025-11-05

**Soundness:** 2
**Presentation:** 2
**Contribution:** 2
**Rating:** 2
**Confidence:** 2

**Summary:**

The paper studies how to preserve Gaussianity in the latent optimization of reward-guided generation. The authors present a constrained optimization approach that directly imposes a Gaussianity constraint on the latent. To introduce project gradient ascent, the authors show an efficient projection update throughout a closed-form projection in spectral domain. Finally, the authors present several experiments to show the performance of the proposed method.

**Strengths:**

- The authors present a constrained optimization approach to impose Gaussianity on the latent prior as Gaussian noise constraints. This is a different perspective compared to previous regularization-based methods.

- The authors analyze Gaussian noise constraints in the spectral domain by showing that the projection can be evaluated explicitly. This is important to ensuring computational efficiency.

- Experiments show the effectiveness of the proposed method in generating realistic images.

**Weaknesses:**

- When maximizing the task-specific reward of latent generative models, it is unclear how preserving Gaussianity for the latent prior affects stable and realistic generation.

- The difference between constraint and regularization is not clearly discussed. For instance, we can always formulate constraints as indicator functions in regularization.

- For the proposed constrained optimization, the feasibility and optimality are not analyzed.

- For preserving Gaussianity, it does not necessarily require Gaussianity for all gradient ascent steps. For instance, we can always project the last step to be a Gaussian.

- Projected gradient ascent can be very expensive, due to the computational cost of projection step and gradient evaluation. Gradient evaluation is not always feasible, for instance reward is non-differentiable.

- It is not discussed how does the inaccurate FFT affect the projection step, since FFT is often computed within some accuracy level.

**Questions:**

See comments in Weaknesses.

---

> ### Author Response · Authors · 2025-11-26
>
> We thank the reviewer for their assessment of our work and for highlighting both strengths and concerns. We respond to each of the raised weaknesses in turn.
> We believe that **all** the listed concerns stem from misunderstandings of our paper or existing techniques, rather than actual weaknesses of our submission. We respectfully ask the reviewer to reconsider their overall assessment of our work.
>
> ---
>
> ### [W1] On how preserving Gaussianity affects stable and realistic generation
>
> We use a one-step generative model (**L65**) that is trained to **map a standard Gaussian sample (white Gaussian noise) to a realistic data point** (**L33–35**). During reward-guided gradient ascent, however, the latent can drift away from this standard Gaussian prior (e.g., in norm and spectral statistics), so the generator is evaluated on *out-of-distribution* latents, which often leads to unstable or unrealistic generations. To prevent this, we can regularize the latent vector to preserve white Gaussian noise characteristics. Please refer to Algorithm 1 for the detailed projected gradient ascent procedure and Section 4.2 for the definition of our white Gaussian noise feasible set.
>
> ---
>
> ### [W2] On the difference between constraints and regularization
>
> In a formal sense, a constraint can be written as an **indicator-function regularizer**. However, this identity does **not** imply that regularization-based methods and our approach behave similarly in practice.
>
> - Regularization methods (prior works) **penalize deviations** from white Gaussian noise characteristics but do not guarantee that **any iterate** satisfies the desired properties; the optimizer can still “trade off” the characteristics against reward.
> - Our method explicitly defines a **white Gaussian noise feasible set** and, crucially, introduces a **closed-form projection** onto this set (highlighted in **L71**, **L143**, **L177**, **L483**). This projection is applied at every step of projected gradient ascent, ensuring that **all iterates remain within the desired feasible set**.
>
> ---
>
> ### [W3] On feasibility and optimality of the constrained optimization
>
> **Feasibility.**
>  - We make the constrained optimization **feasible in practice** by designing constraints that admit a **closed-form projection** in the compact spectral domain. As shown in **Figure 1** and **L463**, one projection takes about **0.47 ms**, which is approximately **0.04%** of the total runtime in our FLUX experiments (50 iterations ≈ 1 minute). In other words, the projection cost is essentially negligible compared to the cost of the generative model and reward evaluation.
>
> **Optimality.**
>  - Guaranteeing global optimality of a nonconvex reward composed with a deep generative model is beyond what any gradient-based method can realistically provide, and we do not claim such guarantees.
>
> If you could elaborate further on what you find insufficient in our analyses of feasibility and optimality, we would be happy to provide additional clarification and details.
>
> ---
>
> ### [W4] On projecting only at the last step vs. at every step
>
> Projecting only at the **final step** amounts to running **unconstrained optimization** and then attempting to “repair” the latent once at the end. By that point, the latent may have moved **far away** from the Gaussian prior. A single final projection must then make a **large correction**, which typically severely damages the target reward.
>
> Our method instead follows standard constrained optimization: **each gradient step** is followed by a projection onto the feasible set, so the entire optimization trajectory **remains in the feasible set**. This is qualitatively different from post-processing and, in our experiments, is crucial to achieving both high rewards and stable, realistic outputs.
>
> ---
>
> ### [W5] On the cost of projected gradient ascent and non-differentiable rewards
>
> In our experiments, the dominant cost comes from:
>
> - the **generative model** forward–backward pass, and
> - the **reward model** forward–backward pass,
>
> which every gradient-based baseline must incur. As noted above, the projection step is extremely cheap (≈0.47 ms, ≈0.04% of runtime, **Figure 1** and **L463**), so the overhead of using projected gradient ascent is **practically negligible** in our setting.
>
> If the reward is non-differentiable, then **all** gradient-based latent optimization methods (including ReNO, PRNO, MPGR, etc.) face the same limitation. Handling non-differentiable rewards is **orthogonal** to our contribution.
>
> ---
>
> ### [W6] On FFT accuracy and its impact on the projection
>
> Our method uses **double-precision FFT/IFFT** implementations from PyTorch. The numerical error of these FFTs is on the order of **10⁻¹⁴**. Given this error level, we do not expect any measurable effect on either the optimization behavior or the generated images.
>
> ---

---

### Author Response · Authors · 2025-12-03
**Summary of the rebuttal discussion**

We would like to summarize the rebuttal discussion. We are genuinely sad that the discussion with the reviewers ended earlier than expected. It would have been very helpful if all reviewers (in particular 7U1f and 738g) had responded to our rebuttal, but only reviewer fY13 engaged with the response and updated the score to 6.

| | Rating | Confidence |
|---|:---:|:---:|
| Reviewer ERkz (no response) | **6** | **4** |
| Reviewer fY13 (responded) | 4 → **6** | **3** |
| Reviewer 738g (no response) | **4** | **3** |
| Reviewer 7U1f (no response) | **2** | **2** |

Overall, the reviewers collectively highlighted the following **strengths** of our work:
- reformulating prior soft regularization approaches as a **hard constrained optimization** framework, and
- demonstrating that this formulation is **practically effective** in our experiments.

Reviewer **ERkz**, who has the highest confidence, even writes that our work “is a valuable contribution that could inspire future work in this direction, … and thus widely applicable.”

Nevertheless, we are particularly disappointed that we could not obtain further responses from reviewers **738g** and **7U1f**, whose weaknesses were based on severe misinterpretations due to an inaccurate or superficial reading:

- **Reviewer 7U1f**: “Projected gradient ascent can be very expensive.”
  **Fact:** In our experiments, *projected* gradient ascent runs in essentially the same time as standard gradient ascent, because the projection step accounts for only **0.04%** of the total runtime. We emphasized this efficient projection (in bold) in Sections 1 and 3, but there was no follow-up discussion.

- **Reviewer 738g**: “The testing dataset’s domain is limited to animals.”
  **Fact:** Our experiments are **not** restricted to animal prompts. We also use the non-animal **T2I-CompBench++** dataset, and, even in our qualitative examples, only **two out of eight** prompts involve animals.

Because these *weaknesses* reflect issues that are not actual limitations of the method, we were very eager to clarify them, but unfortunately the discussion period ended before any further exchange was possible.

For the remaining two reviewers:

- **Reviewer ERkz** raised a single weakness concerning missing references for fundamental facts about Gaussian distributions and the DFT, which could blur the boundary between known facts and our contributions. In the revision, we added the appropriate references and clarified that our contribution lies in **formulating constraints** that leverage these known facts, rather than in re-proving them. This point is now very explicitly addressed, although we did not have the chance to receive further feedback.

- **Reviewer fY13** pointed out three weaknesses regarding **practicality**, **terminology**, and **structure**. We revised the paper to address the terminology (“white Gaussian noise characteristics”) and the presentation, and the reviewer acknowledged these changes as beneficial and **raised their score from 4 to 6**. The only remaining concern is the practicality of test-time optimization, specifically that 60–200 backward passes per prompt may be too costly and that it would be preferable to amortize the cost across multiple prompts. In our follow-up, we clarified that methods applicable to many prompts (fine-tuning–based approaches) are orthogonal to our per-prompt test-time optimization and can be used together, and that our budget of 60-200 forward-backward passes lies in the same range as prior test-time / inference-time scaling methods where users already invest minutes per prompt for high-quality outputs. After this clarification and cost comparison, we could not receive further response.

---

All reviewers agree that our constrained optimization framework is **novel**, **theoretically grounded**, and **practically effective** in our experiments. The concerns regarding references, terminology, and writing structure are all explicitly resolved in the revision. While one concern about the *overall application setting* of test-time optimization remains, we note that there is already a line of work with the same applications published in top-tier conferences. In this context, we believe that our clean and practically effective formulation deserves to be reported in the scholarly literature.

---

### Meta-Review · Area_Chair_8pNL · 2026-01-07

**Summary:**

This paper presents a method for preserving Gaussian noise characteristics latent optimization for a generative model.
The submission received mixed reviews from the reviewers.
The reviewers mainly recognize the idea of using constrained optimization, with sound theoretical grounding and good empirical results.
The main concerns from the reviewers were the efficiency and practicality (7U1f and fY13), necessity of the constraints (7U1f and fY13), terminologies and positioning of paper (fY13), limited evaluation (738g), and missing prior work context (738g and ERkz).
After reading the paper, the reviewers' comments and the authors' rebuttal, the AC believes the authors' responses would have partially addressed the reviewers' concerns, but there would still be outstanding concerns regarding incomplete evaluations and analyses on computation efficiency and behaviors, making the value of the paper much more limiting. The AC believes the merits do not significantly outweigh the weaknesses and does not recommend acceptance at this time.

**Reviewer Concerns:**

Reviewers' concerns mostly addressed:
- Terminologies (fY13)
- Experiments with increased samples and diversity (738g)
- Missing backgrounds (ERkz)
- Sensitivity to block sizes (fY13 and ERkz)

Outstanding concerns:
- Discussions on optimality (7U1f)
- Computation efficiency and practicality (7U1f and fY13)
- Experiments with tasks other than text-to-image (738g)

**Reviewer Scores:**

I think all reviewers would keep their original ratings (with 738g's score raised to 6).

---

### Decision · Program_Chairs · 2026-01-26

Reject